# Local chromatin fiber folding represses transcription and loop extrusion in quiescent cells

**Sarah G Swygert[1]\*, Dejun Lin[2], Stephanie Portillo-Ledesma[3], Po-Yen Lin[4], Dakota R Hunt[1], Cheng-Fu Kao[4], Tamar Schlick[3,5,6], William S Noble[2,7], Toshio Tsukiyama[1]\***

[1]Basic Sciences Division, Fred Hutchinson Cancer Research Center, Seattle, United States; [2]Department of Genome Sciences, University of Washington, Seattle, United States; [3]Department of Chemistry, New York University, New York, United States; [4]Institute of Cellular and Organismic Biology, Academia Sinica, Taipei, Taiwan; [5]Courant Institute of Mathematical Sciences, New York University, New York, United States; [6]New York University-East China Normal University Center for Computational Chemistry at New York University Shanghai, Shanghai, China; [7]Paul G. Allen School of Computer Science and Engineering, University of Washington, Seattle, United States

**Abstract** A longstanding hypothesis is that chromatin fiber folding mediated by interactions between nearby nucleosomes represses transcription. However, it has been difficult to determine the relationship between local chromatin fiber compaction and transcription in cells. Further, global changes in fiber diameters have not been observed, even between interphase and mitotic chromosomes. We show that an increase in the range of local inter-nucleosomal contacts in quiescent yeast drives the compaction of chromatin fibers genome-wide. Unlike actively dividing cells, inter-nucleosomal interactions in quiescent cells require a basic patch in the histone H4 tail. This quiescence-specific fiber folding globally represses transcription and inhibits chromatin loop extrusion by condensin. These results reveal that global changes in chromatin fiber compaction can occur during cell state transitions, and establish physiological roles for local chromatin fiber folding in regulating transcription and chromatin domain formation.

**\*For correspondence:**
sarah.g.swygert@gmail.com (SGS);
ttsukiya@fredhutch.org (TT)

**Competing interest:** The authors declare that no competing interests exist.

## Editor's evaluation

The authors provide compelling evidence that the repression of gene expression during quiescence of the model eukaryote yeast is achieved by heterogenous clustering of local groups of nucleosomes.

## Introduction

The genetic material of eukaryotic cells is organized into a nucleoprotein complex called chromatin, which regulates the accessibility of the underlying sequence to all DNA-dependent processes. The basic unit of chromatin is the nucleosome, 147 base pairs (bp) of DNA wrapped around an octamer of histone proteins containing two copies each of histones H2A, H2B, H3, and H4 (*Kornberg, 1974*; *Luger et al., 1997*; *Davey et al., 2002*). Nucleosomes are arranged on DNA into a 'beads on a string' conformation approximately 10 nanometers (nm) in diameter called the 10 nm fiber (*Kornberg, 1974*; *Davies and Haynes, 1976*; *Finch and Klug, 1976*). The position and occupancy of nucleosomes on the 10 nm fiber are highly regulated and contribute to transcriptional modulation

by impeding transcription factor binding and retarding polymerase elongation (*Rando and Winston, 2012*; *Teves et al., 2014*; *Nocetti and Whitehouse, 2016*). A longstanding hypothesis has been that an even greater level of transcriptional regulation is conferred by the arrangement of 10 nm fibers into three-dimensional higher-order chromatin structures mediated by interactions between nucleosomes located near each other on the DNA strand (*Woodcock et al., 1984*; *Woodcock and Ghosh, 2010*; *Maeshima et al., 2021*; *Luger et al., 1997*; *Dorigo et al., 2004*; *Schalch et al., 2005*). However, mapping chromatin fiber structure at the local nucleosome level in cells and determining its relationship to physiological processes has been technically challenging (*Hsieh et al., 2015*; *Ou et al., 2017*; *Krietenstein and Rando, 2020*).

The most extensive study of chromatin fiber diameter in cells to date used a breakthrough labeling and electron microscopy tomography method, chromEMT, to characterize chromatin in intact human interphase and mitotic cells (*Ou et al., 2017*). This work found that regardless of cell state, chromatin fibers exist in heterogeneous forms 5–24 nm in diameter, suggesting that genome-wide changes in chromatin fiber compaction do not occur even in highly condensed mitotic chromosomes. However, an outstanding question is if gene-to-gene variations in chromatin folding function in transcriptional regulation, or if they are simply a passive byproduct of chromatin dynamics (*Dekker, 2008*; *Krietenstein and Rando, 2020*). Although a genomics method called Micro-C that resolves three-dimensional chromatin structure at single-nucleosome resolution in cells found correlation between the number of inter-nucleosomal interactions within a gene and transcriptional activity in yeast, evidence suggests that changes in chromatin conformation may be a result of or entirely decoupled from transcription (*Hsieh et al., 2015*). Additionally, measurements of inter-nucleosomal contacts conferred by chromatin fiber compaction may be obscured by gene looping in mammals, in which both conformations are expected to increase contacts but to have opposite effects on transcription (*Hsieh et al., 2020*). Similarly, super-resolution imaging and genomics experiments in mammalian B cells suggest that although clutches of folded chromatin fibers called nanodomains decompact upon B cell activation, fiber decompaction requires transcriptional activation (*Kieffer-Kwon et al., 2017*). Collectively, these results demonstrate that the causality relationship between local chromatin fiber conformation and transcription has yet to be firmly established.

Beyond local compaction, chromatin fibers are arranged into large-scale structures called chromatin domains, in which regions of chromatin preferentially interact within other regions in the same domain while being insulated from interactions with other domains (*Dixon et al., 2012*; *Nora et al., 2012*; *Rowley and Corces, 2018*). This organization regulates transcription by promoting enhancer-promoter interactions within domains while inhibiting aberrant enhancer-promoter contacts between domains (*Hnisz et al., 2016a*; *Hnisz et al., 2016b*). A specific form of domain, called a chromatin loop domain, is formed when ring-shaped structural maintenance of chromosomes (SMCs) complexes, such as cohesin or condensin, extrude loops of chromatin that in mammalian cells can be up to megabase size (*Rao et al., 2017*; *Rowley and Corces, 2018*; *Banigan and Mirny, 2020a*). Although chromatin organization is a highly active area of study, how the chromatin fiber contributes to domain structure and whether or not these levels of chromatin architecture cooperate in transcriptional regulation is unknown.

To address these questions, we have used Micro-C to map chromosomal interactions genome-wide at single-nucleosome resolution in purified quiescent *Saccharomyces cerevisiae* (*Hsieh et al., 2015*; *Hsieh et al., 2016*; *Swygert et al., 2019*). Quiescence is a reversible state in which the cell exits the cell cycle in response to external cues and transitions to a long-lived and stress-resistant program (*Gray et al., 2004*; *Coller et al., 2006*; *Cheung and Rando, 2013*). Entry and exit from quiescence are essential for diverse biological processes such as adult stem cell maintenance and lymphocyte activation, and have been linked to drug resistance in pathogenic micro-organisms and cancer recurrence (*Valcourt et al., 2012*; *Rittershaus et al., 2013*; *Chen et al., 2016*). Budding yeast cells enter into quiescence via a coordinated shift in metabolic and gene expression programs in response to glucose exhaustion, and can be purified from non-quiescent cells by density gradient (*Gray et al., 2004*; *Allen et al., 2006*; *Sagot and Laporte, 2019*). Like their mammalian counterparts, quiescent budding yeast undergo dramatic chromatin reorganization and widespread transcriptional reprogramming (*Lohr and Ide, 1979*; *Allen et al., 2006*; *Sagot and Laporte, 2019*), making them an ideal model for determining the mechanisms and functions of three-dimensional chromatin structure in transcriptional regulation.

Given the extensive global level of chromatin condensation in quiescent yeast cells (*Allen et al., 2006*; *Swygert et al., 2019*), we hypothesized that the folding of chromatin fibers contributes to large-scale transcriptional repression during quiescence. We discovered that between cycling (log) and quiescence, an increase in the distance of local inter-nucleosomal interactions drives an increase in the diameter of chromatin fibers. Local nucleosome contacts are fundamentally distinct in log and quiescent cells, with only the latter dependent on a basic patch in the histone H4 tail. Further, quiescence-specific, basic patch-dependent fiber folding globally represses transcription and inhibits chromatin loop extrusion by condensin. These results demonstrate that genome-wide changes in chromatin fiber folding can occur in a physiologically relevant cell state, and that local fiber compaction can play key roles in the regulation of transcription and chromatin loops.

## Results

### Local inter-nucleosomal interactions are distinct between quiescent and actively dividing cells

We have previously shown that distal contacts between nucleosomes over one kilobase (kb) from each other increase in quiescent cells compared to log (*Figure 1—figure supplement 1A*; *Swygert et al., 2019*). This results from the formation of condensin-dependent loops between the boundaries of large chromosomally interacting domains (L-CIDs). To examine differences in the distribution of local inter-nucleosomal interactions, we used our previously published Micro-C XL data to generate maps of contact probabilities of nucleosomes within 1 kb of each other on the DNA strand and to calculate genome-wide odds ratios of long-range interactions between 500 and 1000 bp to short-range interactions between 50 and 500 bp (*Figure 1A*; *Swygert et al., 2019*). Contact probability analysis revealed that, on genome average, the pattern of inter-nucleosomal interactions shifts between log and quiescence. In log, next neighbor (n+1) interactions are strongly favored, with interactions sharply decreasing with increasing distance. The high proportion of n+1 interactions reflects an extended chromatin fiber where nucleosomes are most likely to encounter the nucleosomes closest to them on the DNA strand through random dynamics (*Grigoryev et al., 2009*). In contrast, in quiescent cells, short-range n+1 and n+2 contacts under 500 bp decrease to similar levels, and long-range interactions from 500 to 1000 bp are more frequent. This increase in long-range contacts in quiescent cells is consistent with a locally folded chromatin fiber. To examine the relationship between transcription and local chromatin contacts, we performed hierarchical clustering of our previously published RNA Polymerase II subunit Rpb3 (Pol II) ChIP-seq data in log and quiescent cells to separate Pol II transcribed genes into 'on' and 'off' clusters (*Swygert et al., 2019*). We then generated maps of nucleosome contacts less than 1 kb using these clusters (*Figure 1—figure supplement 1B, C*). This analysis revealed that even relatively highly expressed genes in quiescent cells display an increase in the distance of long-range nucleosome contacts as compared to log, suggesting the increase in long-range contacts is not a consequence of transcriptional repression but instead a genome-wide change in nucleosome contact pattern.

Biochemical experiments using reconstituted components and modeling studies have shown that chromatin fibers with short linker DNA lengths compact into zig-zag conformations with dominant n+2 interactions (*Dorigo et al., 2004*; *Schalch et al., 2005*; *Robinson et al., 2006*; *Collepardo-Guevara and Schlick, 2014*; *Grigoryev et al., 2016*). However, the frequency of n+1 and n+3 interactions in the quiescent cell data is higher than would be expected for a uniform zig-zag conformation. We wondered if this was due to the extensive crosslinking in the Micro-C XL protocol, in which cells are crosslinked with both a short crosslinker (formaldehyde) and a long crosslinker (disuccinimidyl glutarate [DSG]) (*Hsieh et al., 2016*). To test this, we repeated Micro-C XL of log and quiescent cells in the absence of DSG. As previously, and for all Micro-C XL experiments throughout, we used the HiCRep method to verify the agreement between two biological replicates prior to merging replicate data to achieve maximum read depth, then used HiCRep to quantify the differences between conditions (*Figure 1—figure supplement 2A, B*; *Yang et al., 2017*; *Lin et al., 2021*). Omitting DSG only subtly affected Micro-C XL results in quiescent cells, with 200 bp resolution (*Figure 1B*), genome-wide (*Figure 1—figure supplement 3A*), and 1 kb resolution (*Figure 1—figure supplement 3B*) Micro-C XL heatmaps all appearing very similar. Consistently, the HiCRep stratum-adjusted correlation coefficient (SCC) calculated between quiescent cell Micro-C samples with and without DSG was

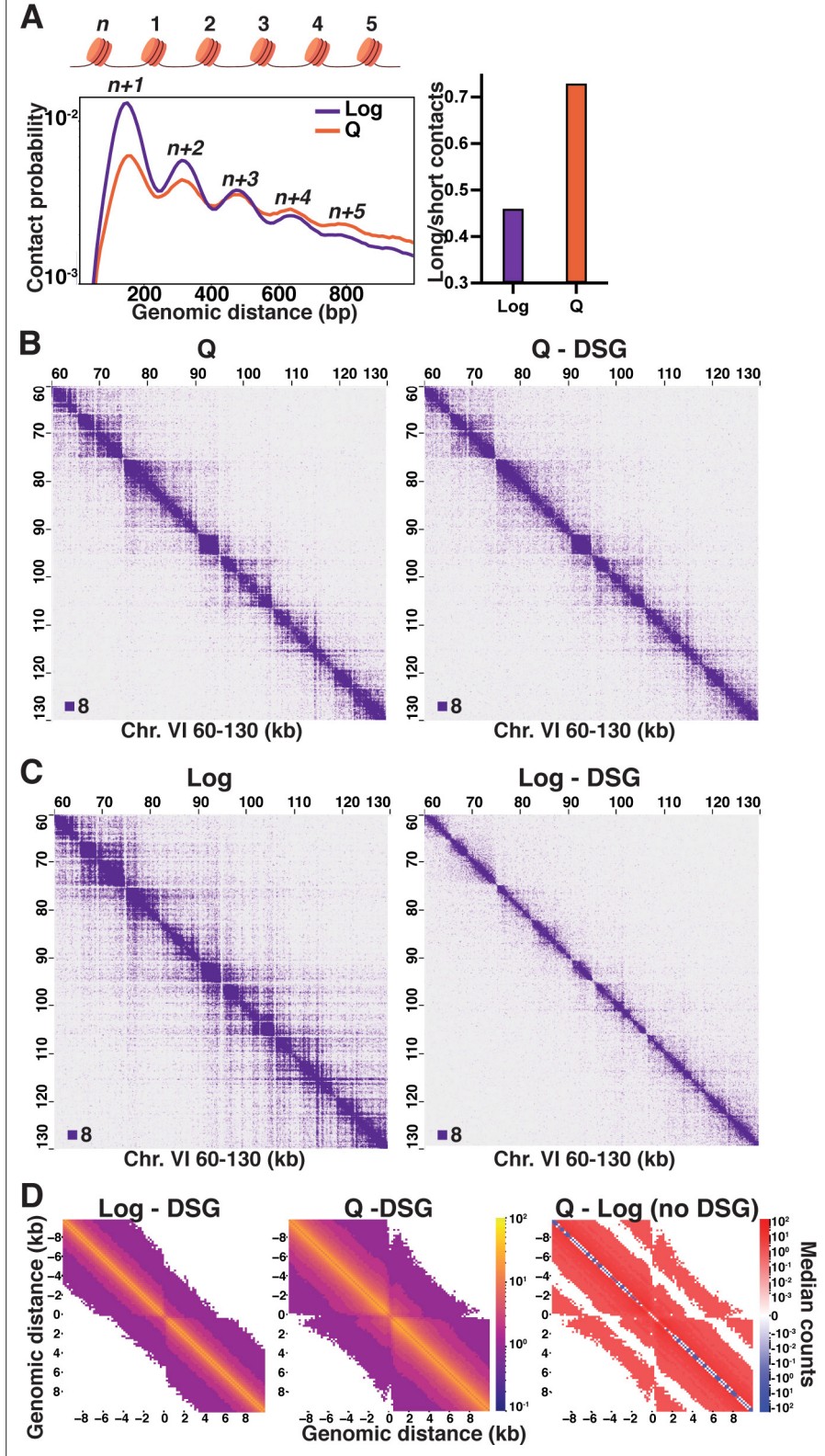

**Figure 1.** Local inter-nucleosome interactions are distinct in quiescent cells. (**A**) Left, contact probability map of nucleosome (n) interactions from exponentially growing (Log) and quiescent (Q) Micro-C XL data (3). Contacts between ligated di-nucleosomes in the 'same' orientation (including 'in-out' and 'out-in' pairs) are shown (4). Data are normalized so that the total probability of intranucleosomal contacts on the same chromosome is equal to 1.

*Figure 1 continued on next page*

*Figure 1 continued*

Right, ratio of contacts between 500 and 1000 bp to contacts between 50 and 500 bp. (**B**) Representative Q and (**C**) Log Micro-C XL with or without DSG at 200 bp resolution. For this and all subsequent Micro-C analyses, all in-facing read pairs were omitted to avoid contamination from undigested di-nucleosomes. (**D**) Micro-C XL metaplots of median interactions ±10 kb around sites of condensin-bound L-CID boundaries at 200 bp resolution. The scale shows the difference between median counts so that contacts in red are increased in Q and contacts in blue are increased in Log. Rightmost plot shows subtraction of the leftmost plot from the center plot.

The online version of this article includes the following figure supplement(s) for figure 1:

**Figure supplement 1.** HiCRep scores of Micro-C data.

**Figure supplement 2.** Long-distance and gene-specific contacts in Log and Q.

**Figure supplement 3.** Omitting DSG from the Micro-C XL protocol diminishes contacts in log cells.

0.95 (*Figure 1—figure supplement 2B*), the same as between biological replicates performed the same way (*Figure 1—figure supplement 2A*). In striking contrast, omitting DSG from Micro-C XL of log cells diminished long-range interactions (*Figure 1C*, *Figure 1—figure supplement 3C, D*), and the SCC between data with and without DSG was 0.82 (*Figure 1—figure supplement 2A, B*). This suggests that use of long crosslinkers disproportionately captures longer-range and/or more transient local inter-nucleosomal interactions in log cells, consistent with a less folded and more dynamic chromatin fiber in log versus quiescent cells. Additionally, genome-wide metaplots generated ±10 kb from condensin-bound L-CID boundaries show that inter-nucleosomal contacts in log cells occur at much lower distance than in quiescent cells (*Figure 1D*). However, although local inter-nucleosomal interactions decreased overall when DSG was omitted, the pattern of distal (*Figure 1—figure supplement 1A*) and nucleosome interactions (*Figure 1—figure supplement 3E*) in log and quiescent cell data remained similar to that with DSG included. These data show that the distribution of inter-nucleosomal contacts dramatically shifts between log and quiescence, from transient, neighboring interactions in log to more stable, longer-range interactions in quiescence that are not consistent with a homogeneous zig-zag conformation.

## Local chromatin fiber compaction increases in quiescent cells

Micro-C data represent the average frequency of inter-nucleosomal interactions within the population of cells used and thus cannot directly report the structure of single molecules. To measure the diameter of individual chromatin fibers, we took a high-angle annular dark-field scanning transmission electron microscopy (HAADF-STEM) tomography approach to image 200 nm cross sections of uranyl acetate and lead citrate stained G1-arrested and quiescent yeast cells (*Figure 2A*). As a positive control, we also imaged magnesium-treated chicken erythrocyte nuclei, one of the few cell types known to undergo extensive chromatin fiber compaction (*Langmore and Schutt, 1980*; *Ou et al., 2017*). STEM tomograms were reconstructed and analyzed to estimate the surface thickness of chromatin fibers (*Figure 2B*). Our results are consistent with chromatin fiber compaction in quiescent versus log cells, with quiescent cell fibers appearing more compact qualitatively and demonstrating an upward shift in diameter as compared to log even at the level of individual chromatin fibers (*Figure 2B and C*). However, quiescent cell fibers were not as compact or as regular as fibers in chicken erythrocyte nuclei (*Figure 2A–C*). We also adopted morphological erosion analysis to estimate chromatin fiber diameter separately (*Figure 2—figure supplement 1A, B*; *Ou et al., 2017*). This found that the average diameter of chromatin fibers in log cell is 15% less than the average chromatin fiber diameter in Q cells, consistent with surface-thickness estimates. These data are consistent with the Micro-C results and demonstrate that chromatin fibers adopt a compact and heterogeneous folded structure in quiescent cells.

To examine chromatin fibers in more detail, we next used our nucleosome-resolution chromatin mesoscale model to investigate the structure of a 40-kb region of the right arm of Chromosome I during log and quiescence (*Beard and Schlick, 2001a*; *Beard and Schlick, 2001b*; *Arya and Schlick, 2006*; *Luque et al., 2014*; *Arya and Schlick, 2009*; *Perišić et al., 2019*; *Portillo-Ledesma and Schlick, 2020*). Following on our previous work on modeling the HOXC and Pou5f1 genes (*Bascom et al., 2019*; *Gómez-García et al., 2021*), we modeled each fiber by incorporating experimentally determined nucleosome positions, H4 tail acetylation, and putative linker histone occupancy (from

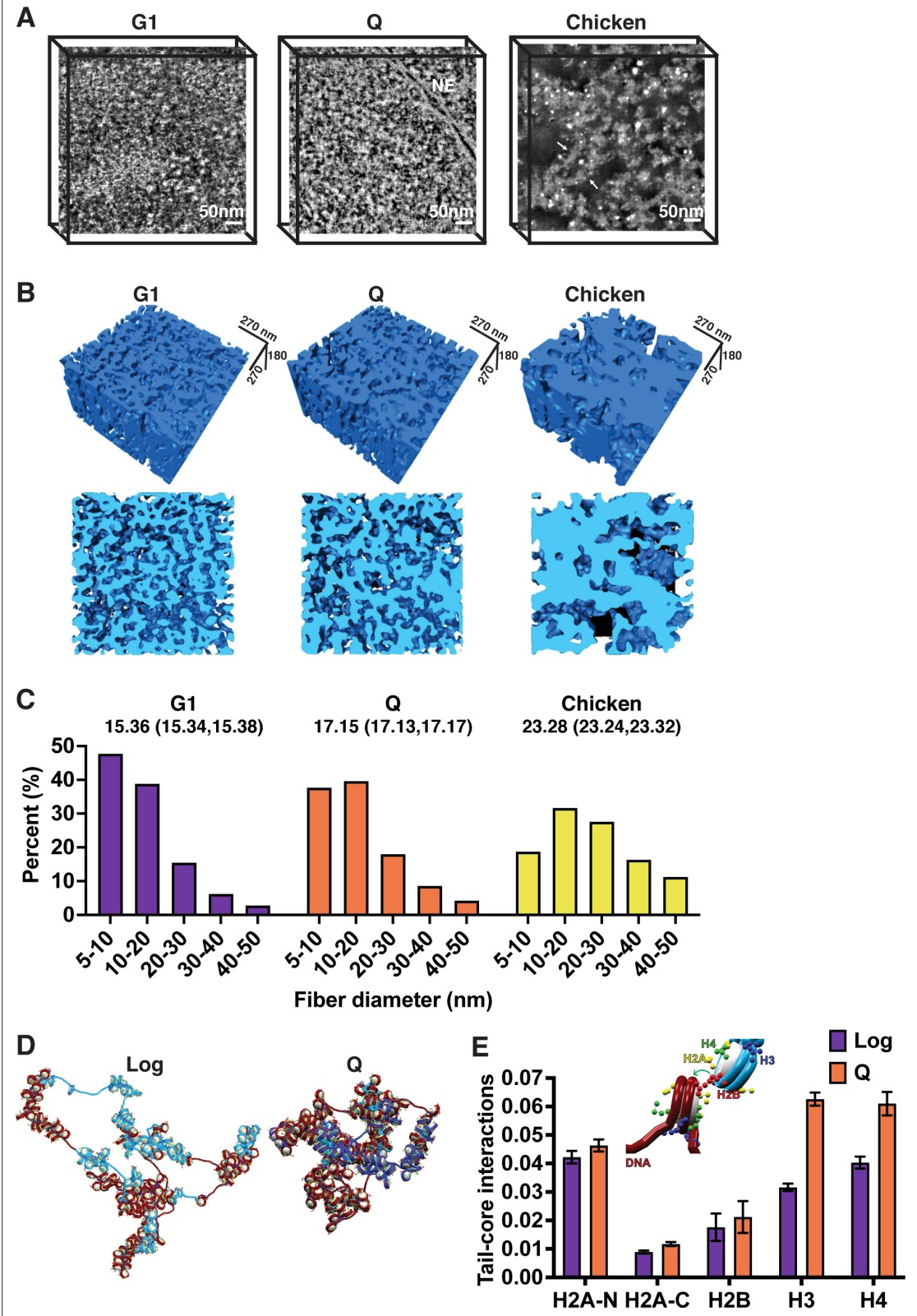

**Figure 2.** Local chromatin fiber compaction increases in quiescent cells. (**A**) HAADF-STEM images of uranyl acetate and lead citrate stained G1-arrested, Q cell, and magnesium-treated chicken erythrocyte nuclei slices. Chromatin fibers appear as white. NE is nuclear envelope. Arrows point to 30 nm fibers. Scale bar represents 50 nm. (**B**) Three-dimensional tomographic reconstructions of yeast and magnesium-treated chicken erythrocyte nuclei. (**C**) Histograms of fiber diameter counts calculated using the surface thickness function in Amira software. Numbers shown are mean diameter

*Figure 2 continued on next page*

*Figure 2 continued*

and 95% confidence interval of the mean. For G1, n=1,028,741. For Q, n=952,372. For chicken erythrocytes, n=316,608. (**D**) Representative equilibrated configurations of Log and Q chromatin fiber models of Chromosome I, 130–170 kb. Genes are shown in blue and intergenic regions are shown in red. Linker histone is shown in turquoise, and histone tails are shown in blue (H3), green (H4), yellow (H2A N- and C-terminal), and red (H2B). (**E**) Normalized contact counts between histone tails and nucleosome cores of separate nucleosomes across 50 trajectories. Here, the normalization of total contacts is determined so that all contacts for each histone tail with other chromatin elements (e.g., DNA beads, nucleosome cores, or other tails) sum up to unity. Refer to the method section 'Tail interaction' for details. Error bars show standard deviation. Inset shows example tail-core contact. ANOVA analysis p-values between log and Q are as follows: H2A-N $6.8e^{-16}$, H2A-C $7.9e^{-43}$, H2B 0.0009, H3 $1.1e^{-92}$, and H4 $5.6e^{-53}$. HAADF-STEM, high-angle annular dark-field scanning transmission electron microscopy.

The online version of this article includes the following figure supplement(s) for figure 2:

**Source data 1.** Tail-core contact counts.

**Figure supplement 1.** Morphological erosion analysis of chromatin fiber diameters.

**Figure supplement 2.** Input parameters for mesoscale modeling.

**Figure supplement 2—source data 1.** Input parameters for modeling.

**Figure supplement 3.** Mesoscale modeling of Log and Q chromatin fibers.

**Figure supplement 3—source data 1.** Compaction parameters.

**Figure supplement 3—source data 2.** Tail-tail contact counts.

**Figure supplement 3—source data 3.** Tail-DNA contact counts.

MNase-seq, histone H4 tail penta-acetylation, and Hho1 ChIP-seq data, respectively) (*Figure 2—figure supplement 2A, B*). As we have shown previously, these parameters are key factors determining fiber compaction and architecture and gene folding (*Perišić et al., 2010*; *Collepardo-Guevara et al., 2015*; *Bascom et al., 2017*; *Bascom and Schlick, 2017*; *Bascom et al., 2019*; *Gómez-García et al., 2021*; *Portillo-Ledesma et al., 2021*). As a result, the two fibers differ by the number of nucleosomes (222 in log vs. 228 in Q), the length of the linker DNAs, the number and length of nucleosome-free regions (*Figure 2—figure supplement 2A, B* and *Figure 2—figure supplement 2—source data 1*), the linker histone density (0.05 in log vs. 0.29 LH/nucleosome in Q), and the histone tail acetylation level (61 in log vs. 3 acetylated nucleosomes in Q). Both systems were simulated by 50 independent trajectories of 60–80 million Monte Carlo steps, carefully monitoring convergence (*Figure 2—figure supplement 3A, B*).

Modeling revealed the same 40 kb region to be more compact in quiescent compared to log cells, with the radius of gyration decreasing and nucleosome packing increasing proportionally (*Figure 2D*, *Figure 2—figure supplement 3C, D*). The modeled contact probability also predicted increased n+1 contacts in log cells and a shift toward longer-range n+2 to n+6 inter-nucleosomal interactions in quiescent cells, though more modest than in experimental data (*Figure 2—figure supplement 3E*). Measurements were also more variable for simulations of the fiber in log, consistent with a more dynamic chromatin fiber in log versus quiescent cells (*Figure 2—figure supplement 3D*). To investigate the mechanisms behind quiescent chromatin folding, we measured the frequency of contacts across configurations between histone tails and the linker DNA of separate nucleosomes (*Figure 2—figure supplement 3F*), histone tails and other histone tails (*Figure 2—figure supplement 3G*), and histone tails and separate nucleosome cores (*Figure 2E*). The frequencies are calculated as the number of chromatin configurations a histone tail is within 2 nm of the indicated nucleosome feature divided by the total number of sampled configurations. H3 and H4 tails demonstrated a striking increase in interactions with the cores of other nucleosomes in quiescence compared to log (*Figure 2E*). These results indicate that quiescent and log cells have fundamentally distinct local chromatin fiber folding.

## Histone deacetylation is necessary for quiescence-specific chromatin folding

We next sought to determine the molecular basis for quiescence-specific local chromatin fiber folding. Biochemical studies have shown that 10 nm chromatin fibers reconstituted from recombinant components compact into a structure approximately 30 nm in diameter, called the 30 nm fiber (*Finch and Klug, 1976*; *Hansen et al., 1989*; *Krietenstein and Rando, 2020*). Folding into 30 nm fibers requires the histone H3 and H4 N-terminal tails, and relies on an interaction between the acidic patch on the surface of a nucleosome and a basic patch in the histone H4 tail of the interacting nucleosome

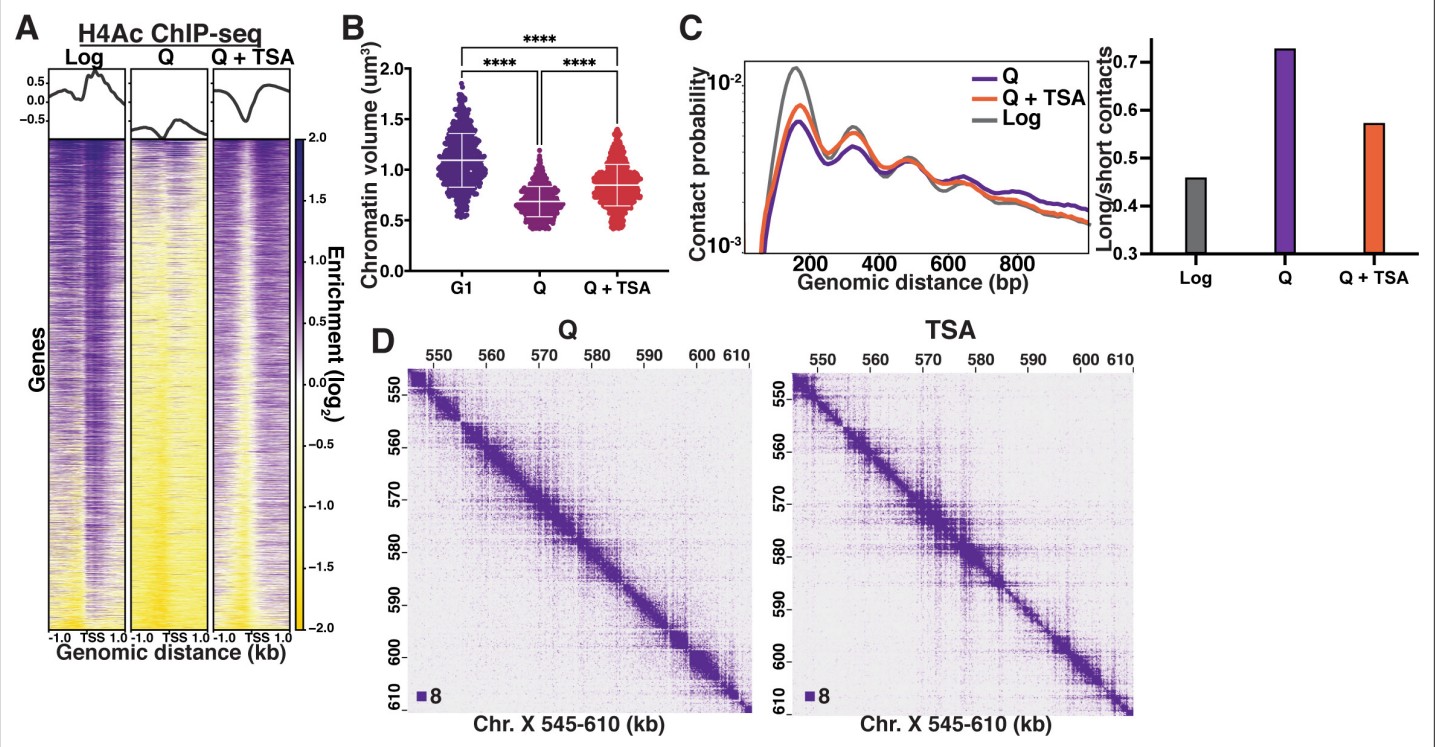

**Figure 3.** Histone deacetylation is necessary for quiescence-specific chromatin folding. (**A**) H4 tail penta-acetylation ChIP-seq heatmaps ±1 kb of all transcription start sites (TSS). Rows are linked across all heatmaps. (**B**) Chromatin volume measurements following DAPI staining of at least 100 cells each of two biological replicates. Bars represent mean and standard deviation. Significance was determined using Welch's ANOVA followed by Games-Howell's multiple comparisons test. Raw data are listed in *Figure 3—source data 1*, and statistics are listed in *Supplementary file 1*. Here, G1 indicates cells arrested in G1. (**C**) Left, contact probability map generated from Micro-C XL data. Data are normalized so that the total probability of intranucleosomal contacts on the same chromosome is equal to 1. Right, ratio of contacts between 500 and 1000 bp to contacts between 50 and 500 bp. (**D**) Representative Micro-C XL data of quiescent cells without (left) and with (right) TSA treatment at 200 bp resolution.

The online version of this article includes the following figure supplement(s) for figure 3:

**Source data 1.** Chromatin volume measurements.

**Figure supplement 1.** TSA treatment increases H4 tail acetylation and decompacts chromatin in Q cells.

**Figure supplement 1—source data 1.** H4Ac and H2B Western blots.

(*Garcia-Ramirez et al., 1992*; *Schwarz et al., 1996*; *Dorigo et al., 2003*; *Dorigo et al., 2004*; *Kan et al., 2009*), which can be disrupted by the acetylation of lysine 16 of the H4 tail (H4K16Ac) (*Shogren-Knaak et al., 2006*). Although both the Micro-C and HAADF-STEM data clearly demonstrate that quiescent cell chromatin does not form 30 nm fibers, we hypothesized that H4 tail deacetylation is involved in quiescence-specific local chromatin fiber folding for two reasons. First, a major difference between log and quiescent chromatin is the massive deacetylation of residues in the histone H3 and H4 tails by the histone deacetylase complex Rpd3 (*McKnight et al., 2015*). Second, as described above (*Figure 2E*), our modeling data show an increase in histone tail-nucleosome core interactions in quiescence. To test this possibility, we first treated cells during a late stage of quiescence entry with the histone deacetylase inhibitor Trichostatin A (TSA), which has previously been shown to disrupt inter-nucleosomal interactions in mammalian cells (*Ricci et al., 2015*; *Otterstrom et al., 2019*). TSA treatment of quiescent cells led to an increase in H4 tail acetylation approaching log cell levels as measured by ChIP-seq (*Figure 3A*), and higher levels than log in bulk as measured by Western blot (*Figure 3—figure supplement 1A*). The ability to restore histone acetylation to log-equivalent levels in quiescent cells by TSA treatment implies that histone acetyltransferases are present and active in the quiescent cell nucleus, and that the global lack of histone acetylation during quiescence results from ongoing deacetylation, just as has been observed in quiescent mammalian B cells (*Kieffer-Kwon et al., 2017*). To determine the effect of histone acetylation on global chromatin condensation in TSA-treated cells, we measured chromatin volume by staining DNA with 4',6-diamidino-2-phenylindole (DAPI) followed

by confocal imaging. As a control, we also imaged DAPI-stained G1-arrested cells. G1 cells were used rather than log for microscopy experiments as they have the same DNA content as quiescent cells. TSA treatment significantly increased the chromatin volume of quiescent cells (*Figure 3B*, *Figure 3— figure supplement 1B*, *Supplementary file 1*). To examine if this change resulted from alterations in local chromatin folding, we next performed Micro-C XL on TSA-treated quiescent cells. Micro-C XL data at 200 bp resolution show that TSA treatment reduces long-range (beyond n+3) and increases short-range (n+1 and n+2) nucleosome contacts (*Figure 3C–D*; see *Figure 3—figure supplement 1C-D* for examples at lower resolution). These results support our model that global histone deacetylation is required for quiescence-specific local chromatin fiber folding, and provide an explanation for why fiber folding is distinct in log and quiescence.

## The H4 tail basic patch regulates quiescence-specific chromatin folding

We next sought to determine if histone deacetylation is necessary for quiescence-specific chromatin fiber folding due to obligate interactions between the nucleosome acidic patch and the H4 basic patch. To this end, we created yeast strains in which the endogenous H3 and H4 loci were deleted and complemented by a mutant or wild-type (WT) copy of H3 and H4 genes at an ectopic locus. The WT control strains (HHF2) grow and enter quiescence very similarly to true WT strains with two copies of H3 and H4 genes. DAPI staining and volume measurements of quiescent cells show that alanine and glutamine substitutions of K16 (K16A and K16Q) significantly increase quiescent cell chromatin volume to a similar extent (*Figure 4A*, *Figure 4—figure supplement 1A*, *Supplementary file 1*). Substitution of both arginine residues with alanine (R17R19A) decompacts chromatin even further, and full abrogation of all five basic patch residues ([16]KRHRK[20]) with alanine (5toA) decompacts chromatin almost to G1 levels. In contrast, during G1, neither R17R19A nor 5toA strains displayed decompaction compared to HHF2 (*Figure 4B*, *Supplementary file 1*). Consistent with the basic patch playing an important role specifically in quiescence, strains bearing H4 basic patch substitutions do not display altered growth in log, but have up to an 80% reduction in quiescence entry (*Figure 4—figure supplement 1B*), although cells that did enter maintained similar longevity to WT (*Figure 4—figure supplement 1C*). Substitutions to H2A residues in the acidic patch by the same genetic strategy similarly increased chromatin volume (*Figure 4—figure supplement 1D*, *Supplementary file 1*), supporting the model that H4 tail/acidic patch contacts may be involved in chromatin folding in quiescent cells. However, we did not pursue further experiments with these H2A mutant strains because they exhibited stronger growth defects and were more difficult to work with than the H4 mutants.

We next performed Micro-C XL experiments using quiescent cells of the H4 mutants displaying the most chromatin decompaction, R17R19A and 5toA, as well as of the control strain, HHF2. HHF2 was indistinguishable from WT (*Figure 1—figure supplement 2B*). In contrast, both basic patch mutants displayed strong chromatin fiber decompaction at the local level in quiescent cells (*Figure 4C–E*). Other than telomere clustering defects evident in genome-wide plots as previously shown (*Laporte et al., 2016*), distal interactions displayed less prominent differences from WT (*Figure 4—figure supplement 2A-E*). However, consistent with massive chromatin unfolding, the pattern of local inter-nucleosomal interactions dramatically shifted, with long-range interactions beyond n+2 strongly decreasing in both mutants, and short-range n+1 interactions increasing relative to n+2 (*Figure 4F*). Metaplots centered around L-CID boundaries similarly showed a large reduction in local chromatin contacts as compared to WT (*Figure 4G*). Collectively, these experiments support a model in which local chromatin fiber folding in quiescent cells is mediated by inter-nucleosomal interactions driven by the H4 basic patch.

## Quiescence-specific local chromatin folding represses transcription

The ability to disrupt quiescence-specific chromatin fiber folding through H4 mutation gave us the opportunity to determine the role of this compaction in transcriptional regulation during quiescence. To this end, we performed ChIP-seq of Pol II in log and quiescent mutant cells. We have previously shown that Rpb3 ChIP-seq correlates well with ChIP-seq of phosphorylated serine 2 of the Pol II carboxy-terminal domain and is thus an appropriate way to measure active transcription (*Swygert and Tsukiyama, 2019*; *Swygert et al., 2019*), while avoiding complications introduced in RNA measurements by the storage of mRNAs in stress granules and processing bodies during quiescence (*Yamasaki and Anderson, 2008*; *Li et al., 2013*; *Swygert et al., 2019*). Consistent with previous

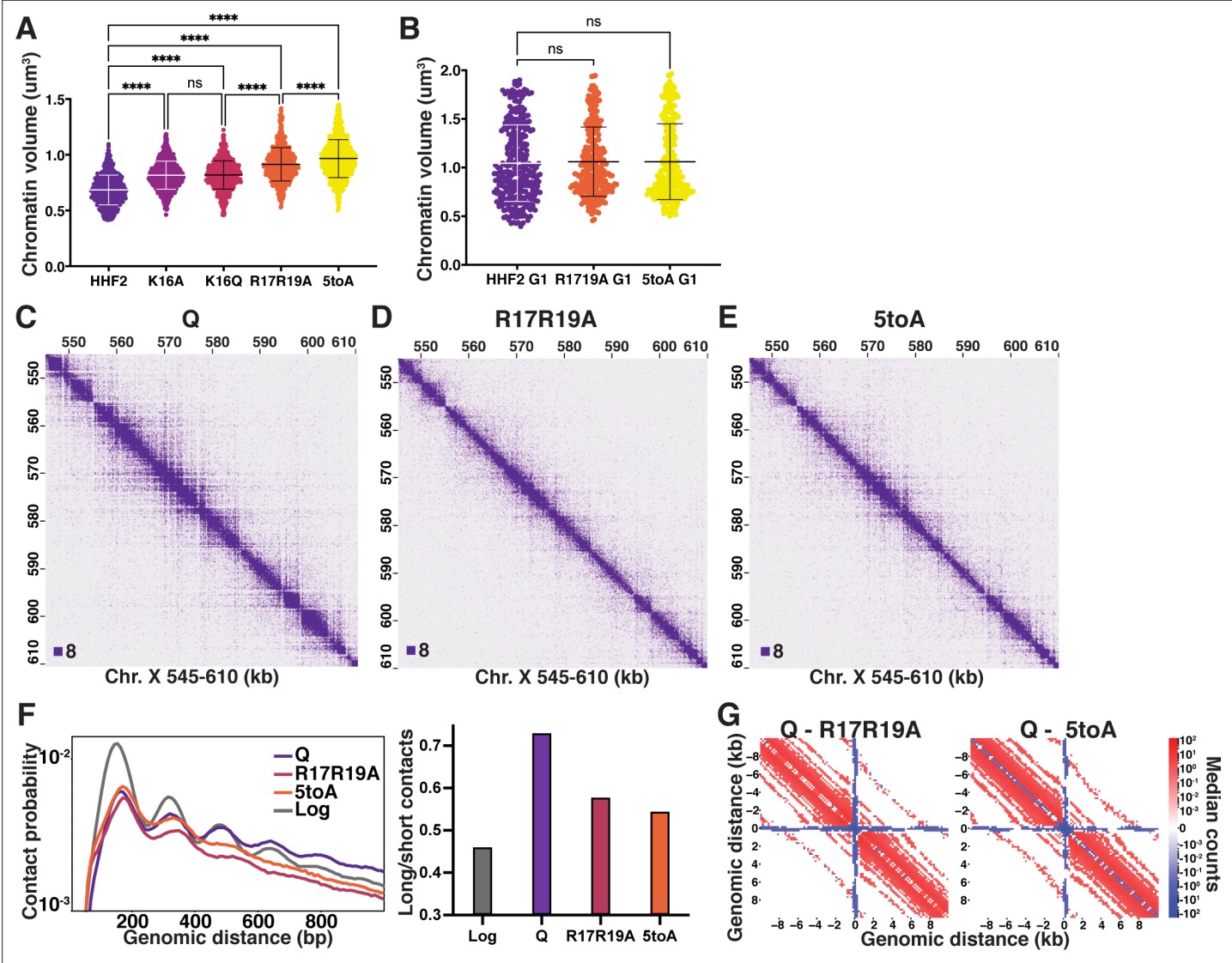

**Figure 4.** The H4 tail basic patch regulates quiescence-specific chromatin folding. (**A**) Chromatin volume measurements following DAPI staining of at least 100 cells of two biological replicates of quiescent H4 mutant cells. HHF2 is the single-copy HHT2-HHF2 WT control strain. Bars represent mean and standard deviation. Significance was determined using Welch's ANOVA followed by Games-Howell's multiple comparisons test. Raw data are listed in *Figure 4—source data 1*, and statistics are listed in *Supplementary file 1*. (**B**) Chromatin volume measurements as in (**A**) of HHF2 and H4 mutant G1 cells following DAPI staining of two biological replicates. G1 cells were selected by morphology as H4 mutant strains cannot be arrested in G1. Raw data are listed in *Figure 4—source data 1*, and statistics are listed in *Supplementary file 1*. (**C–E**) Representative Micro-C XL data of WT (**C**) and H4 mutant (**D–E**) quiescent cells at 200 bp resolution. (**F**) Left, contact probability map generated from Micro-C XL data. Q and Log indicate cells in the indicated stage from true WT strains. Data are normalized so that the total probability of intranucleosomal contacts on the same chromosome is equal to 1. Right, ratio of contacts between 500 and 1000 bp to contacts between 50 and 500 bp. (**G**) Subtraction Micro-C XL metaplots of median interactions around sites of condensin-bound L-CID boundaries in quiescent cells at 200 bp resolution. The scale shows the difference between median counts so that contacts in red are increased in WT Q and contacts in blue are increased in the mutants.

The online version of this article includes the following figure supplement(s) for figure 4:

**Source data 1.** Chromatin volume measurements.

**Figure supplement 1.** Phenotypes of H4 basic patch and H2A acidic patch substitutions in quiescence.

**Figure supplement 1—source data 1.** Q cell counts.

**Figure supplement 1—source data 2.** Q cell longevity.

**Figure supplement 1—source data 3.** Chromatin volume measurements.

**Figure supplement 2.** H4 basic patch substitutions decompact chromatin in Q.

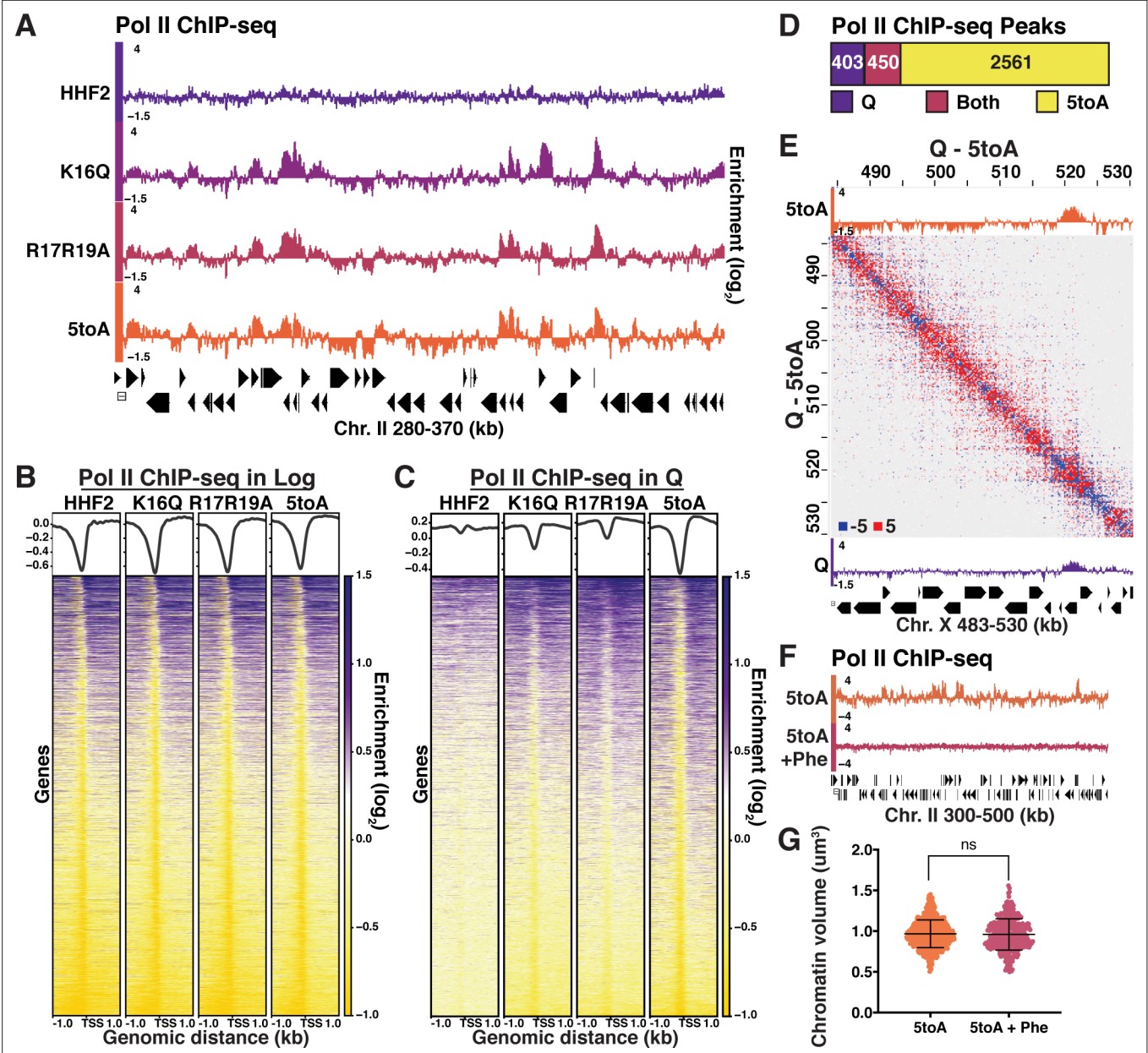

**Figure 5.** Quiescence-specific local chromatin folding represses transcription. (**A**) Genome browser view of Pol II subunit Rpb3 ChIP-seq data in quiescent mutant strains across a portion of Chromosome II. (**B**) Heatmaps of Rpb3 across all TSSs in Log and (**C**) Q cells. Rows are the same across all heatmaps in a panel. (**D**) MACS differential peak calls for WT and 5toA Rpb3 ChIP-seq in Q. (**E**) Pol II ChIP-seq data in 5toA (top) and true WT Q (bottom) cells overlaid on a representative heatmap showing 5toA Micro-C XL data subtracted from WT Q data. Positive (red) indicates contacts that are higher in WT cells. (**F**) Genome browser view of Rpb3 ChIP-seq data in 5toA quiescent cells with or without 1,10-phenanthroline treatment (5toA+Phe). (**G**) Chromatin volume measurements following DAPI staining of at least 100 cells each of two biological replicates. Bars represent mean and standard deviation. Significance was determined using a two-tailed unpaired t-test with Welch's correction. Exact numbers are listed in the Materials and methods. TSS, transcription start site.

The online version of this article includes the following figure supplement(s) for figure 5:

**Source data 1.** Chromatin volume measurements.

**Figure supplement 1.** Pol II and H3 ChIP-seq in Log and Q.

results (*Fazzio et al., 2005*), genome-browser tracks (*Figure 5A*, *Figure 5—figure supplement 1A*), and Pearson correlation scores (*Figure 5—figure supplement 1B*) displayed minimal differences in Pol II occupancy between basic patch mutants and HHF2 in log. In contrast, Pol II occupancy increased dramatically in quiescent basic patch mutant cells, with occupancy increasing proportionally to the number of substitutions introduced to the basic patch (*Figure 5A–C*, *Figure 5—figure supplement 1A, C*). ChIP-seq peak calling implemented using MACS2 similarly showed a dramatic increase in Pol II peaks between HHF2 and 5toA quiescent cells, with the number of peaks increasing in the 5toA mutant corresponding to approximately 40% of all genes (*Figure 5D*).

Although these results strongly support the model that local chromatin fiber folding represses global transcription in quiescent cells, another possible interpretation of these data is that transcriptional activation leads to chromatin fiber decompaction. However, activation appears to be downstream of chromatin decompaction, as fiber unfolding occurs in the basic patch mutants genome-wide, regardless of the transcriptional state of the underlying gene (*Figure 5E*, *Figure 5—figure supplement 1D-F*). This suggests that local chromatin folding represses transcription, but unfolding is not sufficient for activation. To test this idea more directly, we treated 5toA cells with the transcription inhibitor 1,10-phenanthroline during quiescence entry. Phenanthroline treatment leads to complete loss of Pol II occupancy by ChIP-seq (*Figure 5F*). However, DAPI staining of phenanthroline-treated 5toA quiescent cells shows no change in chromatin volume (*Figure 5G*, *Supplementary file 1*), supporting that chromatin decompaction in 5toA cells is a cause and not a result of global transcriptional activation.

## H4 tail-mediated chromatin folding inhibits condensin loop extrusion

Although the role of SMC complexes in chromatin domains has been a highly active subject of exploration, the relationship between domain formation and the conformation of the underlying chromatin fiber is not understood (*Banigan and Mirny, 2020a*). We previously found that in quiescent yeast, condensin relocates from its positions in log cells to form chromatin loop domains called L-CIDs, whose boundaries are at the promoters of coding genes (*Swygert et al., 2019*). To determine if chromatin fiber compaction affects condensin loop domain formation during quiescence, we performed ChIP-seq of the condensin subunit Brn1 in the H4 basic patch mutant strains. By overlaying the ChIP-seq and Micro-C data, we identified the striking appearance of stripes that overlapped with condensin subunit Brn1 ChIP-seq peaks in WT quiescent cells (*Figures 6A and 2G*, *Figure 6—figure supplement 1A-C*). Stripes are believed to occur in Hi-C data as a result of loop extrusion, in which SMC complexes bind the genome at two points and increase the size of the resulting loop by moving chromatin in one or both directions through the SMC ring (*Goloborodko et al., 2016*; *Vian et al., 2018*; *Banigan et al., 2020b*). In the case of one-sided extrusion, chromatin at the fixed boundary progressively contacts chromatin that is being extruded, leading to the formation of a stripe when contacts are measured at the population level (*Figure 6B*). Although stripes are not observable at all Brn1 sites, aggregate peak metaplots generated from WT, R17R19A, and 5toA Micro-C data showing *trans* interactions between Brn1 peaks show clear stripe patterns at the mean and median in the mutants (*Figure 6C*, *Figure 6—figure supplement 1D*). Consequently, our Micro-C results suggest that substitutions in the H4 tail affect the way in which condensin extrudes chromatin loops.

Changes in loop extrusion would also be expected to result in alterations in condensin binding. Although bulk Brn1 protein levels did not appreciably change between the mutants and HHF2 (*Figure 6—figure supplement 1E*), Brn1 binding patterns were altered in the mutants (*Figure 6D* and *Figure 6—figure supplement 1F, G*). While Brn1 shows strong localization just upstream of transcription start sites in WT and HHF2 quiescent cells, basic patch mutants show a reduction in the magnitude of Brn1 peaks and a corresponding spreading out across gene bodies on either side of promoters (*Figure 6D*) and L-CID boundaries (*Figure 6E*). MACS peak calling further showed a decrease in the number of Brn1 peaks between HHF2 and 5toA (*Figure 6F*), which we interpret to mean that Brn1 spreading across the genome reduces many discrete peaks to below detectable levels. Overlaying Brn1 ChIP-seq tracks on the Micro-C XL data shows that the flattening and broadening of Brn1 peaks are easily observable at the level of individual stripe loci (*Figure 6A*, *Figure 6—figure supplement 1A-C*). This shows that stripes at least in part result from progressive Brn1 binding to regions of chromatin within L-CIDs, as would be expected from active loop extrusion.

Modeling studies have predicted that the emergence of stripes in Hi-C data is likely to result from an increase in the rate at which SMC complexes extrude chromatin loops: as loop extrusion occurs

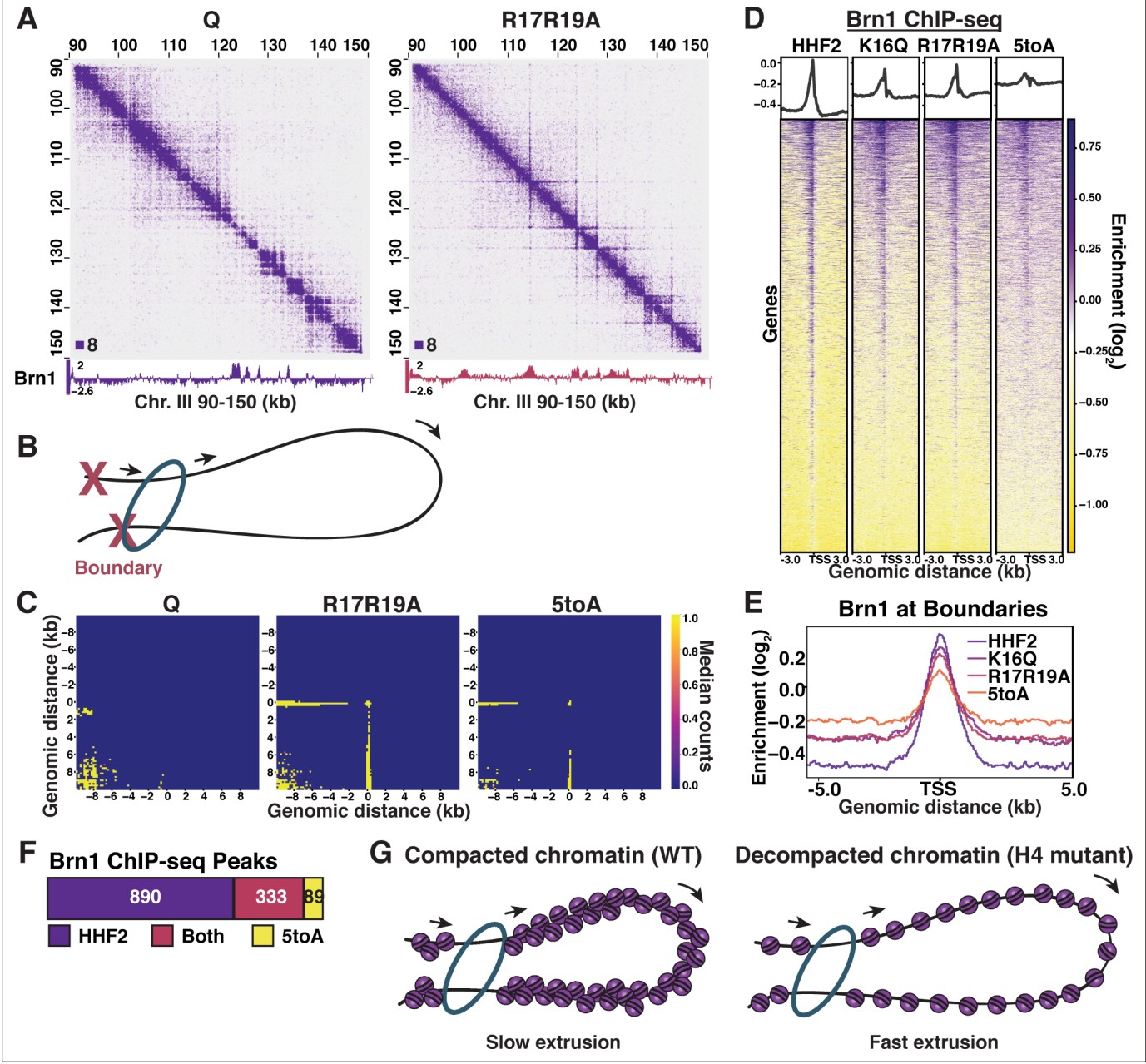

**Figure 6.** H4-tail mediated chromatin folding inhibits condensin loop extrusion. (**A**) Condensin subunit Brn1 ChIP-seq data overlayed beneath representative Micro-C XL data at 200 bp resolution. (**B**) Schematic showing one-sided loop extrusion by condensin between two boundaries. (**C**) Metaplots of Micro-C XL data showing aggregate peak analysis of *trans* nucleosome contacts ±10 kb between Brn1 ChIP-seq peaks in the indicated condition. Data shown are the median contact values. (**D**) Heatmaps of Brn1 ChIP-seq ±3 kb of all TSSs. Rows are the same across all heatmaps. (**E**) Metaplot of Brn1 ChIP-seq data ±5 kb of L-CID boundaries. (**F**) MACS differential peak calls for HHF2 and 5toA Brn1 ChIP-seq in Q. (**G**) Model: In WT quiescent cells, local nucleosome interactions drive the compaction of chromatin fibers. This compaction inhibits condensin loop extrusion, resulting in slow or paused extrusion. In H4 mutant quiescent cells, loss of nucleosome interactions leads to fiber decompaction and an increased rate of condensin loop extrusion. TSS, transcription start site; WT, wild-type.

The online version of this article includes the following figure supplement(s) for figure 6:

**Figure supplement 1.** H4-mediated chromatin fiber folding represses condensin loop extrusion.

**Figure supplement 1—source data 1.** Brn1-FLAG and H2B Western blots.

more rapidly, interacting regions of chromatin have less time to diffuse apart and consequently retain 'memory' of being extruded (***Nuebler et al., 2018***). Additionally, constant rapid extrusion is likely

to be captured at all points of passage at the cell population level. Although transcription has been proposed to affect loop extrusion through multiple avenues—by slowing the process of extrusion by creating difficult-to-traverse transcription bubbles, by 'pushing' SMC complexes across chromatin, and by promoting SMC complex loading—our data do not support that the change in condensin extrusion we see in the H4 mutants is due to the increase in transcription in the mutants (*Lengronne et al., 2004*; *Busslinger et al., 2017*; *Tran et al., 2017*; *Zhang, 2020*). This is because: (1) stripes occur uniformly between L-CID boundaries, regardless of Pol II occupancy (as seen in *Figure 6A*, *Figure 6—figure supplement 1A-C*); (2) stripes appear as strong in the R17R19A mutant as in the 5toA mutant, despite less Pol II occupancy overall; and (3) regions displaying stripes do not exhibit large changes in Pol II occupancy between WT and the mutants, because Brn1 sites tend to overlap with Pol II-occupied genes even in WT (*Figure 6—figure supplement 1G*; *Swygert et al., 2019*). Instead, we propose a model in which H4 tail-mediated chromatin fiber folding sterically impedes the ability of condensin to extrude chromatin, leading to slower loop extrusion than in mutants with decompacted fibers (*Figure 6G*). This slower extrusion may help to stabilize loops between L-CID boundaries in WT quiescent cells.

## Chromatin fiber folding and condensin looping are complementary mechanisms of transcriptional repression

We previously found that condensin depletion during quiescence entry de-represses about 20% of all genes, with genes within 1 kb of a condensin binding site disproportionately represented (*Swygert et al., 2019*). To compare the extent of transcriptional repression conferred by chromatin fiber compaction versus chromatin loop domain formation, we used MACS2 to determine the difference in Pol II peaks in quiescent cells in which the condensin subunit SMC4 was conditionally depleted using a tet-off system (*SMC4*-off) (*Swygert et al., 2019*) and quiescent cells of the 5toA mutant. While many Pol II peaks appear in both *SMC4*-off and 5toA cells, there are large numbers of non-overlapping Pol II peaks, with nearly double the number of distinct peaks in the 5toA mutant (*Figure 7A*). As previously shown (*Swygert et al., 2019*), genes with the highest Pol II occupancy in *SMC4*-off cells tend to be close to Brn1 sites, while Pol II occupancy in 5toA cells does not necessarily overlap with Brn1 (*Figure 7B*). Additionally, although condensin depletion appears to disrupt repression by decreasing insulation at L-CID boundaries and decreasing local inter-nucleosomal interactions to some extent compared to WT (*Swygert et al., 2019*), the H4 mutants display significantly more fiber decompaction than *SMC4*-off cells, while retaining some insulation at boundaries (*Figure 7C*). Interestingly, although short-range n+1 and n+2 nucleosome contacts increase in *SMC4*-off cells compared to WT, long-range (500–1000 bp) contacts do not decrease as dramatically as in the H4 mutants, resulting in similar long/short contact odds ratios (*Figure 7D*). These data suggest that condensin affects local contacts differently and to a lesser extent than the H4 tail, and that the inhibition of condensin loop extrusion is a complementary rather than primary mechanism of transcriptional repression by H4 tail-mediated chromatin fiber compaction. Further, while condensin is largely responsible for regulating the transcription of genes located around condensin binding sites at L-CID boundaries during quiescence, H4-mediated chromatin compaction regulates genes across the genome.

## Discussion

Although compaction into regular 30 nm fibers is a robust property of biochemically reconstituted chromatin, similar structures have not been identified in the majority of cell types (*Maeshima et al., 2019*). Instead, physiological chromatin fibers are irregularly folded into structures with a broad range of diameters, with most falling below 30 nm in size (*Dekker, 2008*; *Ou et al., 2017*). It has previously been unclear if variations in chromatin fiber compaction are functional or simply a byproduct of dynamic DNA-dependent processes (*Krietenstein and Rando, 2020*). Our data show that between log and quiescence, an increase in the range of local inter-nucleosomal interactions compacts chromatin fibers genome-wide. These fibers are less compact and entirely distinct from 30 nm fibers, as they lack the expected increase in n+2 interactions seen in extended zig-zag 30 nm conformations by population average (Micro-C), and are smaller in diameter at the individual fiber level (as determined HAADF-STEM). Instead, quiescent cell fibers display an overall decrease in n+1 and n+2 contacts compared to cycling cells in favor of longer-range inter-nucleosomal interactions beyond 500 bp,

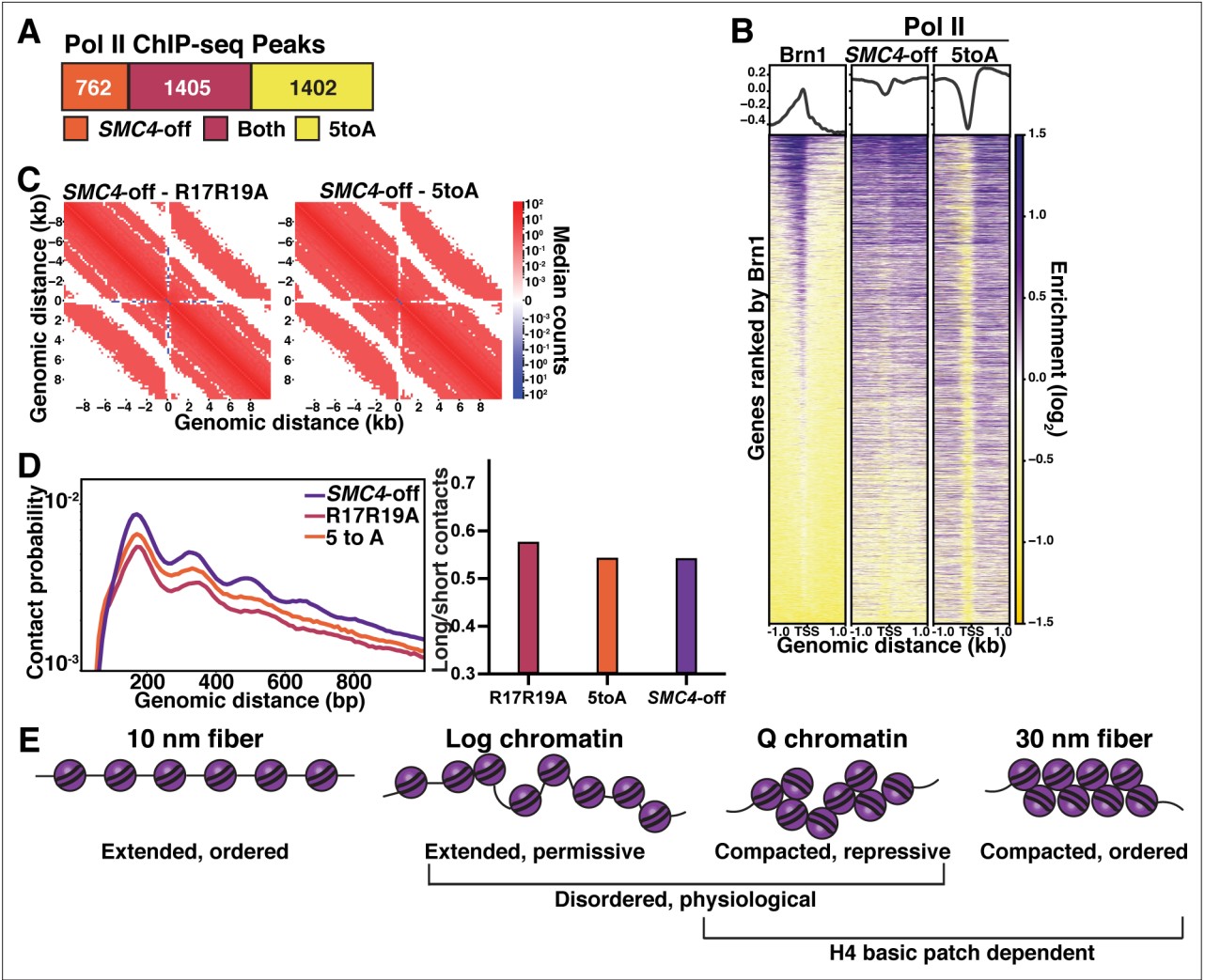

**Figure 7.** Chromatin fiber folding and condensin looping are complementary mechanisms of transcriptional repression. (**A**) MACS differential peak calls for *SMC4*-off and 5toA Brn1 ChIP-seq in Q. *SMC4*-off cells contain a doxycycline-inducible Tet repressor system to shut off the expression of the condensin subunit SMC4. *SMC4*-off data were previously published in *Swygert et al., 2019*. (**B**) Heatmaps of Brn1 and Pol II (subunit Rbp3) ChIP-seq ±1 kb of all TSSs. All heatmaps are ordered by descending Brn1 occupancy in WT Q (leftmost heatmap). (**C**) Micro-C XL subtraction metaplots of median interactions around sites of condensin-bound L-CID boundaries in quiescent cells at 200 bp resolution. The scale shows the difference between median counts. Plots show mutant Micro-C XL data subtracted from previously published *SMC4*-off Micro-C XL data (*Swygert et al., 2019*). (**D**) Left, contact probability map of Micro-C XL data. Data are normalized so that the total probability of intranucleosomal contacts on the same chromosome is equal to 1. Right, ratio of contacts between 500 and 1000 bp to contacts between 50 and 500 bp. (**E**) Model of physiologically relevant local chromatin fiber compaction compared to canonical chromatin fiber models. In log, chromatin fibers are in a disordered, extended state that is permissive to transcription. In Q, an increase in local nucleosomal interactions driven by H4 deacetylation promotes folding into disordered, compacted structures that are transcriptionally repressive. TSS, transcription start site.

consistent with extensive but disordered compaction into a more heterogeneous structure. Despite their differences in inter-nucleosomal interaction pattern, however, quiescent cell fibers, like 30 nm fibers, rely on the H4 tail basic patch, suggesting that disordered structures in quiescent cells and compacted ordered reconstituted structures rely on a shared mechanism. We propose that although physiological fibers lack regular structure, most likely due to chromatin dynamics, the irregularity of nucleosome spacing and occupancy, and the binding of additional architectural factors (*van Holde and Zlatanova, 2007*), extended chromatin fibers in log cells and compacted fibers in quiescent cells represent physiological versions of distinct fiber folding states (*Figure 7E*).

We previously found that the histone deacetylase complex Rpd3 is targeted across the genome during quiescence entry and massively deacetylates histone H3 and H4 tails (*McKnight et al., 2015*). Loss of

Rpd3 leads to defects in quiescence entry and quiescent cell longevity, and prevents global transcriptional repression. Similarly, mammalian quiescent cells exhibit widespread histone deacetylation (*Rawlings et al., 2011*; *Evertts et al., 2013*; *Kieffer-Kwon et al., 2017*). Consequently, our current results suggest a conserved mechanism for the role of histone deacetylases in regulating transcription during quiescence. They also establish the role of H4K16Ac in local chromatin fiber folding and transcriptional regulation in cells. Consistently, H4K16Ac is commonly implicated in mediating transitions in state. For example, H4K16Ac increases the most of all chromatin modifications upon B-cell activation (*Kieffer-Kwon et al., 2017*), bookmarks genes for expression during zygotic genome activation in flies (*Samata et al., 2020*), and disrupts dosage compensation (*Wells et al., 2012*; *Copur et al., 2018*).

Although it has generally been hypothesized that chromatin compaction is refractory to transcription through the steric hindrance of transcription factor binding and polymerase elongation, a causal relationship between compaction and transcriptional inhibition has been difficult to establish in cells. Our data show that local chromatin fiber compaction mediated by the H4 basic patch drives transcriptional repression in quiescent cells. This result is in contrast to findings in quiescent mammalian B cells, in which histone acetylation by TSA treatment leads to an expansion in chromatin volume, but chromatin nanodomains mediated by inter-nucleosomal interactions do not decompact in the absence of transcriptional activation and a resulting increase in ATP production by Myc (*Kieffer-Kwon et al., 2017*). It is unclear if these discrepancies reflect differences between yeast and mammalian quiescence, in which mammalian cells may employ additional mechanisms of chromatin fiber compaction, or if chromatin nanodomains monitored by super-resolution microscopy are distinct from chromatin fiber compaction detected by Micro-C. Importantly, our results show that TSA treatment does not decompact chromatin as extensively as abrogation of the basic patch, so it is likely that chromatin decompaction upon quiescence exit requires pathways in addition to H4K16Ac. This likely also explains differences between our results and findings in quiescent fission yeast, in which TSA treatment does not upregulate transcription (*Cai, 2020*). In this case, we would expect transcriptional levels upon TSA treatment to change modestly, similar to K16Q mutants.

In addition to chromatin fiber decompaction, our results show that H4 basic patch mutation accelerates condensin loop extrusion. We propose that during quiescence entry, condensin forms L-CID loops by the process of extrusion (*Nasmyth, 2001*; *Goloborodko et al., 2016*; *Banigan and Mirny, 2020a*). As chromatin fibers compact, the rate of extrusion decreases or stalls, resulting in persisting loops but not stripes in quiescent cells. Support for this model comes from quiescent murine B cells, in which existing cohesin loops can persist for hours in the absence of ATP, and cohesin stripes increase on B cell activation (*Vian et al., 2018*). Interestingly, although yeast condensin has been shown to extrude loops in one direction in vitro (*Ganji et al., 2018*), we see that stripes and condensin spreading emanate from boundaries in both directions. These data support a model in which multiple condensins bind L-CID boundaries and extrude loops in opposite directions, possibly through a version of the handcuff model in which an anti-parallel dimer of condensins is loaded at boundaries (*Banigan and Mirny, 2020c*). Alternately, multiple condensin monomers may be loaded at boundaries in random orientation, resulting in bi-directional extrusion, particularly when viewed at the cell population level (*Banigan et al., 2020b*; *Sebastian Jimenez, 2021*). Further, although the CTCF-dependent mechanism of loop domain boundary formation present in mammalian cells is not conserved in yeast, stripes can be observed extending across multiple L-CID boundaries in yeast just as they do in mammals (*Vian et al., 2018*). This suggests that, as in mammalian cells, yeast boundaries are not always sufficient to block SMC complex extrusion, perhaps through boundary bypass or the absence of boundary elements in a subset of cells in the population.

Collectively, our data show that local chromatin fiber contacts can fundamentally change during physiologically relevant conditions, and that contacts consistent with local chromatin fiber folding can repress transcription and loop extrusion. In the future, more work is needed to determine the relationship between fiber compaction and loop extrusion, as well as to uncover additional mechanisms driving changes in chromatin and transcription between log and quiescent cells. Additionally, as nucleosome-resolution chromatin mapping tools become more widely available, we expect that local chromatin compaction will be demonstrated to regulate processes in a variety of contexts and cell types.

# Materials and methods

## Key resources table

| Reagent type (species) or resource | Designation | Source or reference | Identifiers | Additional information |
|---|---|---|---|---|
| Strain, strain background (*Saccharomyces cerevisiae*) | WT | *McKnight et al., 2015* | yTT5781 | MATa RAD5+ prototroph: |
| Strain, strain background (*S. cerevisiae*) | WT | *McKnight et al., 2015* | yTT5783 | MATa RAD5+ prototroph |
| Strain, strain background (*S. cerevisiae*) | Hho1-FLAG | This Paper | yTT6336 | MATa RAD5+ prototroph HHO1-2L-3FLAG::KanMX |
| Strain, strain background (*S. cerevisiae*) | Hho1-FLAG | This Paper | yTT6337 | MATa RAD5+ prototroph HHO1-2L-3FLAG::KanMX |
| Strain, strain background (*S. cerevisiae*) | HHF2 | This Paper | yTT7177 | MATa RAD5+ ura3-1 hht1-hhf1::Nat hht2-hhf2::Hyg trp1-1::pRS 404-HHT2-HHF2 |
| Strain, strain background (*S. cerevisiae*) | HHF2 | This Paper | yTT7206 | MATa RAD5+ ura3-1 hht1-hhf1::Nat hht2-hhf2::Hyg trp1-1::pRS 404-HHT2-HHF2 |
| Strain, strain background (*S. cerevisiae*) | hhf2-5toA | This Paper | yTT7175 | MATa RAD5+ ura3-1 hht1-hhf1::Nat hht2-hhf2::Hyg trp1-1::pRS 404-HHT2-hhf2-K16A, R17A,H18A,R19A,K20A |
| Strain, strain background (*S. cerevisiae*) | hhf2-5toA | This Paper | yTT7208 | MATa RAD5+ ura3-1 hht1-hhf1::Nat hht2-hhf2::Hyg trp1-1::pRS 404-HHT2-hhf2-K16A, R17A,H18A,R19A,K20A |
| Strain, strain background (*S. cerevisiae*) | hhf2-R17A,R19A | This Paper | yTT7200 | MATa RAD5+ ura3-1 hht1-hhf1::Nat hht2-hhf2::Hyg trp1-1::pRS 404-HHT2-hhf2-R17A,R19A |
| Strain, strain background (*S. cerevisiae*) | hhf2-R17A,R19A: | This Paper | yTT7207 | MATa RAD5+ ura3-1 hht1-hhf1::Nat hht2-hhf2::Hyg trp1-1::pRS 404-HHT2-hhf2-R17A, R19A |
| Strain, strain background (*S. cerevisiae*) | hhf2-K16A | This Paper | yTT7202 | MATa RAD5+ ura3-1 hht1-hhf1::Nat hht2-hhf2::Hy g trp1-1::pRS404-HHT2-hhf2-K16A |
| Strain, strain background (*S. cerevisiae*) | hhf2-K16A | This Paper | yTT7209 | MATa RAD5+ ura3-1 hht1-hhf1::Nat hht2-hhf2::Hyg trp1-1::pRS404-HHT2-hhf2-K16A |
| Strain, strain background (*S. cerevisiae*) | hhf2-K16Q | This Paper | yTT7205 | MATa RAD5+ ura3-1 hht1-hhf1::Nat hht2-hhf2::Hyg trp1-1::pRS 404-HHT2-hhf2-K16Q |
| Strain, strain background (*S. cerevisiae*) | hhf2-K16Q | This Paper | yTT7210 | MATa RAD5+ ura3-1 hht1-hhf1::Nat hht2-hhf2::Hyg trp1-1::pRS404-HHT2-hhf2-K16Q |

*Continued on next page*

*Continued*

| Reagent type (species) or resource | Designation | Source or reference | Identifiers | Additional information |
|---|---|---|---|---|
| Strain, strain background (*S. cerevisiae*) | HHF2 Brn1-FLAG | This Paper | yTT7390 | MATa RAD5+ ura3-1 hht1-hhf1::Nat hht2-hhf2::Hyg trp1-1::pRS 404-HHT2-HHF2 Brn1-2L-3FLAG::KanMX |
| Strain, strain background (*S. cerevisiae*) | HHF2 Brn1-FLAG | This Paper | yTT7391 | MATa RAD5+ ura3-1 hht1-hhf1::Nat hht2-hhf2::Hyg trp1-1::pRS 404-HHT2-HHF2 Brn1-2L-3FLAG::KanMX |
| Strain, strain background (*S. cerevisiae*) | hhf2-5toA Brn1-FLAG | This Paper | yTT7388 | MATa RAD5+ ura3-1 hht1-hhf1::Nat hht2-hhf2::Hyg trp1-1::pRS 404-HHT2-hhf2-K16A, R17A,H18A,R19A, K20A Brn1-2L-3FLAG::KanMX |
| Strain, strain background (*S. cerevisiae*) | hhf2-5toA Brn1-FLAG | This Paper | yTT7389 | MATa RAD5+ ura3-1 hht1-hhf1::Nat hht2-hhf2::Hyg trp1-1::pRS 404-HHT2-hhf2-K16A, R17A,H18A,R19A,K20A Brn1-2L-3FLAG::KanMX |
| Strain, strain background (*S. cerevisiae*) | hhf2-R17A,R19A Brn1-FLAG | This Paper | yTT7392 | MATa RAD5+ ura3-1 hht1-hhf1::Nat hht2-hhf2::Hyg trp1-1::pRS 404-HHT2-hhf2-R17A, R19A Brn1-2L-3FLAG::KanMX |
| Strain, strain background (*S. cerevisiae*) | hhf2-R17A,R19A Brn1-FLAG | This Paper | yTT7393 | MATa RAD5+ ura3-1 hht1-hhf1::Nat hht2-hhf2::Hyg trp1-1::pRS 404-HHT2-hhf2-R17A, R19A Brn1-2L-3FLAG::KanMX |
| Strain, strain background (*S. cerevisiae*) | hhf2-K16Q Brn1-FLAG | This Paper | yTT7394 | MATa RAD5+ ura3-1 hht1-hhf1::Nat hht2-hhf2::Hyg trp1-1::pRS404-HHT2-hhf2-K16Q Brn1-2L-3FLAG::KanMX |
| Strain, strain background (*S. cerevisiae*) | hhf2-K16Q Brn1-FLAG | This Paper | yTT7395 | MATa RAD5+ ura3-1 hht1-hhf1::Nat hht2-hhf2::Hyg trp1-1::pRS 404-HHT2-hhf2-K16Q Brn1-2L-3FLAG::KanMX |
| Strain, strain background (*S. cerevisiae*) | HTA1 | This Paper | yTT6767 | MATa W303 Rad5+ ade2-1 can1-100 his3-11,15 leu2-3,112 ura3-1 hta1-htb1::Hyg hta2-htb2::Nat trp1-1::pRS 404-HTA1-HTB1 |
| Strain, strain background (*S. cerevisiae*) | HTA1 | This Paper | yTT6773 | MATa W303 Rad5+ ade2-1 can1-100 his3-11,15 leu2-3,112 ura3-1 hta1-htb1::Hyg hta2-htb2::Nat trp1-1::pRS 404-HTA1-HTB1 |

*Continued on next page*

*Continued*

| Reagent type (species) or resource | Designation | Source or reference | Identifiers | Additional information |
|---|---|---|---|---|
| Strain, strain background (*S. cerevisiae*) | hta1-E57A | This Paper | yTT6768 | MATa W303 Rad5+ ade2-1 can1-100 his3-11,15 leu2-3,112 ura3-1 hta1-htb1::Hyg hta2-htb2::Nat trp1-1::pRS404-hta1-E57A-HTB1 |
| Strain, strain background (*S. cerevisiae*) | hta1-E57A | This Paper | yTT6774 | MATa W303 Rad5+ ade2-1 can1-100 his3-11, 15 leu2-3,112 ura3-1 hta1-htb1::Hyg hta2-htb2::Nat trp1-1::pRS404-hta1-E57A-HTB1 |
| Strain, strain background (*S. cerevisiae*) | hta1-E65A | This Paper | yTT6769 | MATa W303 Rad5+ ade2-1 can1-100 his3-11,15 leu2-3,112 ura3-1 hta1-htb1::Hyg hta2-htb2::Nat trp1-1::pRS404-hta1-E65A-HTB1 |
| Strain, strain background (*S. cerevisiae*) | hta1-E65A | This Paper | yTT6776 | MATa W303 Rad5+ ade2-1 can1-100 his3-11,15 leu2-3,112 ura3-1 hta1-htb1::Hyg hta2-htb2::Nat trp1-1::pRS404-hta1-E65A-HTB1 |
| Strain, strain background (*S. cerevisiae*) | hta1-E93A | This Paper | yTT6772 | MATa W303 Rad5+ ade2-1 can1-100 his3-11,15 leu2-3,112 ura3-1 hta1-htb1::Hyg hta2-htb2::Nat trp1-1::pRS404-hta1-E93A-HTB1 |
| Strain, strain background (*S. cerevisiae*) | hta1-E93A | This Paper | yTT6779 | MATa W303 Rad5+ ade2-1 can1-100 his3-11,15 leu2-3,112 ura3-1 hta1-htb1::Hyg hta2-htb2::Nat trp1-1::pRS 404-hta1-E93A-HTB1 |
| Strain, strain background (*S. cerevisiae*) | hta1-E65A,D92A | This Paper | yTT6618 | MATa W303 Rad5+ ade2-1 can1-100 his3-11,15 leu2-3,112 ura3-1 hta1-htb1::Hyg hta2-htb2::Nat trp1-1::pRS 404-hta1-E65A, D92A-HTB1 |
| Strain, strain background (*S. cerevisiae*) | hta1-E65A,D92A | This Paper | yTT6765 | MATa W303 Rad5+ ade2-1 can1-100 his3-11, 15 leu2-3,112 ura3-1 hta1-htb1::Hyg hta2-htb2::Nat trp1-1::pRS404-hta1-E65A,D92A-HTB1 |
| Antibody | H2B (polyclonal, rabbit) | Active Motif | 39237 | WB (1:5000) |
| Antibody | H3 (rabbit polyclonal) | Abcam | 1791 | ChIP (1 µl) |
| Antibody | H4 penta-acetyl (rabbit polyclonal) | MilliporeSigma | 06-946 | WB (1:1000) ChIP (2 µl) |
| Antibody | Rpb3 (mouse monoclonal) | BioLegend | 665003 | ChIP (2 µl) |
| Antibody | FLAG M2 (mouse monoclonal) | Sigma-Aldrich | F1804 | WB (1:3000) ChIP (4 µl) |

*Continued on next page*

*Continued*

| Reagent type (species) or resource | Designation | Source or reference | Identifiers | Additional information |
|---|---|---|---|---|
| Chemical compound | Percoll | GE | 17-0891-01 | |
| Chemical compound, drug | Trichostatin A (TSA) | TCI | T247710MG | |
| Chemical compound, drug | Disuccinimidyl glutarate (DSG) | Thermo Fisher Scientific | PI20593 | |
| Commercial assay or kit | MinElute PCR Cleanup Kit | Qiagen | 28004 | |
| Other | Dynabeads M-280 sheep anti-mouse IgG beads | Invitrogen | 11201D | 20 µl |
| Other | Dynabeads Protein G beads | Invitrogen | 10004D | 20 µl |
| Commercial assay or kit | Ovation Ultralow v2 Kit | Tecan | 0344 | |
| Other | PTFE printed slides | Electron Microscopy Sciences | 63430-04 | |

## Yeast growth and quiescent cell purification

Quiescent cells were grown as previously described (*Allen et al., 2006*; *Spain et al., 2018*), by using single colonies to inoculate cultures in rich media in flasks with at least a 1:5 ratio of culture to flask volume. Cells were grown for 7 days, pelleted, and resuspended in 2.5 ml of water prior to loading on density gradients. Gradients were prepared using 25 ml of a mixture containing 90% Percoll (GE, catalog #17-0891-01) and 150 mM NaCl in 50 ml high-speed round bottom centrifuge tubes. Gradients were centrifuged at 10,000×*g* for 15 min prior to loading samples, then centrifuged at 300×*g* for 1 hr. Quiescent cells were removed from the bottom of gradients by pipetting, washed in water, and quantified using spectrophotometry. For TSA-treatment, TSA (TCI, catalog # T247710MG) was resuspended to 50 mM in DMSO, and added to cultures to 50 µM 24 hr prior to purifying quiescent cells. During purification, 50 µM TSA was also added to Percoll gradients and water was used to resuspend cells. For 1,10-phenanthroline treatment, phenanthroline was dissolved in methanol to 100 mg/ml, then added to cultures for a final concentration of 150 ug/ml 24 hr prior to purifying quiescent cells. For G1-arrested cells, cultures were inoculated to an optical density at $A_{660nm}$ of 0.06, then α-factor was added to 10 µg/ml once the optical density reached 0.15. Cultures were monitored for G1 arrest under the microscope until at least 95% appeared to be in G1, approximately 90 min later. For DAPI staining measurements of H4 mutants in G1, WT and mutant G1 cells were selected from a mixed culture of cells in log based on morphological appearance under bright-field view prior to chromatin volume measurement.

## Micro-C XL

For quiescent cells, 2400 optical density units of purified cells were resuspended in 1.2 L water and crosslinked using 106.8 ml 37% formaldehyde for 10 min at 30°C with shaking. Formaldehyde was attenuated with 120 ml 2.5 M glycine, and cells were pelleted, washed, and resuspended in 120 ml Buffer Z (1 M sorbitol, 50 mM Tris pH 7.4, and 10 mM β-mercaptoethanol). Cells were split into twelve 15 ml conical tubes and spheroplasted by addition of 1 ml 10 mg/ml 100T zymolyase at 30°C with rotation until at least 60% of cells appeared as spheroplasts under the microscope (approximately 45–120 min). Spheroplasts were centrifuged for 10 min at 3000 rpm and 4°C, washed in cold phosphate-buffered saline (PBS), and pelleted again. For DSG crosslinking, DSG (Thermo Fisher Scientific, #PI20593) was resuspended to 300 mM in DMSO and diluted to 3 mM in room temperature PBS. Spheroplasts were resuspended in 5 ml DSG solution and crosslinked for 40 min at 30°C with rotation. Crosslinking was quenched by the addition of 1 ml 2.5 M glycine, and crosslinked spheroplasts were pelleted, washed in cold PBS, and pelleted again prior to flash freezing. Log cells were prepared as above, except six 100 ml cultures were grown to optical density of 0.55 /ml. Cells were spheroplasted in six conical tubes each of cells in 10 ml Buffer Z using 250 µl 10 mg/ml 20T zymolyase at 30°C with rotation for 30 min. For Micro-C XL in the absence of DSG, cells were prepared as above except they were flash-frozen after formaldehyde crosslinking and again following spheroplasting.

For all experiments, two conical tubes of frozen prepared spheroplasts were split into eight MNase titration reactions to determine the concentration giving approximately 95% mononucleosome-sized fragments in the insoluble chromatin fraction, and this concentration of MNase was then used to carry out MNase digestion of the remaining sample. For all experiments, MNase digestion, end repair and labeling, proximity ligation, di-nucleosomal DNA purification, and library preparation were then carried out in 10 reactions for quiescent cells and 4 reactions for log cells exactly as described in *Hsieh et al., 2016*, except that during library preparation, adapter ligation was completed overnight at room temperature. We found that this small modification increases the yield of unique di-nucleosome fragments in our hands. Purified di-nucleosomal DNA following the gel extraction step was combined from all 10 reactions for quiescent cells and all four for log prior to the library amplification. Micro-C XL samples were completed in two biological replicates at different times, agreement between replicates was determined by HiCRep (see below and extended data *Figure 1*), and then replicate sequencing data were merged. Experiments were completed to obtain a minimum of 80 million unique paired reads per merged sample following removal of rDNA and read-through di-nucleosome reads.

## ChIP-seq

### Chromatin preparation

For log cells, approximately 70 optical density units (at $A_{660nm}$) of cells in 100 ml rich media were cross-linked with 3 ml 37% formaldehyde for 20 min with rotation at room temperature, formaldehyde was attenuated with 3.3 ml 2.5 M glycine for 5 min, and cells were washed in cold TBS and flash-frozen. For quiescent cells, approximately 300 optical density units of purified cells were resuspended in 25 ml water and crosslinked with 750 µl 37% formaldehyde as above, then attenuated using 1.25 ml glycine. Pellets were resuspended in 300 µl ice-cold Breaking Buffer (100 mM Tris pH 8, 20% glycerol, and 1 mM PMSF), approximately 600 µl of acid-washed glass beads were added, and cells were bead beat for 5 min (log cells) or 10 min (quiescent cells) until greater than 95% of cells were visibly broken under the microscope. Lysates were separated from beads, centrifuged at maximum speed for 1 min, then pellet was resuspended in 1 ml ice-cold FA Buffer (50 mM HEPES-KOH pH. 7.6, 150 mM NaCl, 1 mM EDTA, 1% TritonX-100, 0.1% sodium deoxycholate, and 1% PMSF). Chromatin was sonicated until fragmented to about 200 bp, then centrifuged two times at 16,000×*g* for 10 min at 4°C to remove residual insoluble material, and flash-frozen and stored at − 80°C. For quantification, an aliquot of chromatin was resuspended in an equal volume of Stop Buffer (20 mM Tris pH 8, 100 mM NaCl, 20 mM EDTA, and 1% SDS) and incubated at 65°C overnight to remove crosslinks. Chromatin was then treated with 0.2 mg/ml RNaseA for 1 hr at 42°C and 0.2 mg/ml Proteinase K for 4 hr at 55°C, purified using the Qiagen MinElute PCR Cleanup Kit (catalog #28004), and quantified using a Qubit system.

### ChIP-seq

Antibodies were conjugated to 20 µl per sample of magnetic beads with shaking for 1 hr at 20°C. For H3 ChIP, 1 µl of anti-H3 antibody (Abcam, catalog #1791) was conjugated to Dynabeads Protein G beads (Invitrogen, catalog #10004D) per reaction. For Pol II ChIP, 2 µl anti-Rpb3 antibody (BioLegend, catalog #665003) was conjugated to Dynabeads M-280 sheep anti-mouse IgG beads (Invitrogen, catalog #11201D). For FLAG ChIP-seq (for Brn1 and Hho1), 4 µl of anti-FLAG antibody (Sigma-Aldrich, #F1804) was conjugated to Protein G beads, and for penta-H4 acetylation ChIP, 2 µl anti-penta-H4Ac antibody (Sigma-Aldrich, #06-946) was conjugated to Protein G beads. Beads were washed in PBST, then added to 1 µg of chromatin. Chromatin was incubated with beads with rotation at room temperature for 1.5 hr, then beads were washed three times in FA Buffer, three times in FA-High Salt Buffer (50 mM HEPES-KOH pH. 7.6, 500 mM NaCl, 1 mM EDTA, 1% TritonX-100, and 0.1% sodium deoxycholate), and one time in RIPA Buffer (10 mM Tris pH 8, 0.25 M LiCl, 0.5% IGEPAL, 0.5% sodium deoxycholate, and 1 mM EDTA). To elute, 50 µl of Stop Buffer was added and beads were incubated at 75°C for 15 min, twice. Elutions were combined and crosslinks were reversed overnight at 65°C. Chromatin was treated with RNase and PK and cleaned up as described. ChIP-seq libraries were prepared using the Ovation Ultralow v2 Kit (Tecan, catalog #0344). Single-end sequencing was completed on an Illumina HiSeq 2500 in rapid run mode. ChIP experiments were completed in two biological replicates.

## Microscopy and chromatin volume measurements

### DAPI staining and confocal microscopy

G1 cells were used in place of log for microscopy experiments in order to ensure cells with the same DNA content as quiescent cells were compared during chromatin volume measurements. For DAPI staining, 5 optical density units at A660nm of cells were fixed in 1 ml of 3.7% formaldehyde in 0.1 M KPO4 pH 6.4 at 4°C for 20 min with rotation. Cells were washed once in 0.1 M KPO4 pH 6.4, re-suspended in 1 ml of sorbitol/citrate (1.2 M sorbitol, 100 mM K2HPO4, and 36.4 mM citric acid). Cells were then digested at 30°C in 200 µl of sorbitol/citrate containing 0.25 µg (for G1) or 2.5 µg (for quiescent) of 100T zymolyase for 5 min (log) or 30 min (quiescent). Cells were washed once and resuspended in sorbitol/citrate, then loaded onto PTFE printed slides from Electron Microscopy Sciences (catalog #63430-04) coated with 0.1% polylysine. Slides were incubated in ice-cold methanol for 3 min and ice-cold acetone for 10 s, before air-drying. Then 10 µl of DAPI-mount (0.1 µg/ml DAPI, 9.25 mM p-phenylenediamine, dissolved in PBS, and 90% glycerol) was added to each slide well. Z stack images with a 0.1-µm interval were obtained using a Leica TCS SP8 confocal microscope at 630× and optimized resolution.

### Nuclear chromatin volume measurement and statistical analysis

Nuclear volumes were measured using the 3D Objects Counter tool in Fiji (*Schindelin et al., 2012*). Threshold was set to best cover the nuclear chromatin of a maximum intensity Z-Projection. Outliers were defined as any data points less than the lower bound (Q1 −1.5 IQR) or greater than the upper bound (Q3 +1.5 IQR.) Significance was determined using either Welch's ANOVA followed by Games-Howell's multiple comparisons test or unpaired t-test with Welch's correction as specified in he figure legend. Statistical tests were conducted using GraphPad Prism 9 and results are listed in *Supplementary file 1*.

## Quiescent cell longevity assay

Purified quiescent cells from 1-week-old cultures were inoculated into water at an optical density ($A_{660nm}$) of 0.1 and incubated at 30°C with constant rotation. Samples were diluted into distilled water before plating onto YPD plates in triplicate. Plates were incubated at 30°C for 2 days before colony counting. Survival was determined by colony forming units (CFUs). CFU at the first week was set to be the initial survival (100%).

## STEM sample preparation and imaging

Chicken erythrocytes were prepared as described previously (*Langmore and Schutt, 1980*). Briefly, erythrocytes were washed in wash buffer (130 mM NaCl, 5 mM KCl, and 10 mM HEPES, pH 7.3) and centrifuged into pelleted cells. Nuclei were prepared by resuspension of pelleted cells in lysis buffer (15 mM NaCl, 60 mM KCl, and 15 mM HEPES, pH 7.3, 0.1% NP-40, and 1 µM PMSF) for 5 min and centrifuged at 1000×*g* for 3 min, resuspension and wash was repeated three times. Nuclei in lysis buffer were then treated with 2 mM MgCl2 and fixed with 2.5% glutaraldehyde and 4% paraformaldehyde at 4°C, postfixed in 1% OsO4, dehydrated in series concentrations of ethanol and embedded in Spurr's resin.

Yeast cells in 20% BSA were cryofixed using High Pressure Freezer (HPM101, Leica) and transferred to freeze-substitution machine (AFS2, Leica). The frozen cells were freeze-substituted in the presence of 2% osmium tetroxide and 0.1% uranyl acetate in acetone at –78°C for 48 hr, warmed up to –20°C for 12 hr followed by 8 hr at room temperature. The fixative was washed out with acetone and embedded in Spurr's resin at room temperature (*Giddings et al., 2001*).

Resin embedded cells were sectioned into either 90 nm sections for TEM/STEM imaging or 200 nm sections for STEM tomogram acquisition and mounted on copper grids. Sections were double-stained with uranyl acetate and lead citrate at room temperature for enhancing contrast before EM acquisition. TEM and STEM imaging were performed using a FEI Tecnai microscope operated at 120 kV with a 2k by 2k CCD camera (US1000, Gatan) and a high-angle annular dark field (HAADF) detector (Model 3000, Fischione), respectively.

For STEM tomography, the 200 nm sections were immersed in a solution of 10 nm colloidal gold particles served as fiducial markers with 1% BSA for 30 ss and air-dried. To reduce the missing wedge effect, the grids were loaded in a rotation sample holder (Model 2040, Fischione) for tomogram

acquisition and the dual axial tilt images were collected from –60° to 60° at 2° increments. The STEM tomograms were reconstructed by simultaneous iterative reconstruction technique with 10 iterations implemented in IMOD software (*Mastronarde, 1997*). Image segmentation processing was performed in Fiji, including contrast enhancement, image binary, and noise removal. The segmented EM sub-tomogram volumes (270 nm×270 nm×180 nm) were imported into Amira software (Thermo Fisher Scientific) to calculate chromatin fiber diameter using the surface thickness function (*Ou et al., 2017*).

For morphological erosion analysis, the EM sub-tomogram volumes were imported into Fiji and continuous change in the radius of the structural element in the Gray morphology function was used to generate a serial erosion volume. The residual volume fraction is calculated by the ratio of the chromatin fiber for each radius to the total chromatin fiber space in a tomogram. Last, the average diameter of the chromatin fiber is the x-axis intercept estimated by the linear fit of the first five data points.

## Mesoscale modeling

### Mesoscale model

Chromatin fibers typical of Log and quiescent cells were modeled using our nucleosome resolution mesoscale model (*Arya and Schlick, 2006*; *Arya and Schlick, 2009*; *Perišić et al., 2010*; *Collepardo-Guevara and Schlick, 2014*). Our chromatin mesoscale model combines four coarse-grained elements: the nucleosome core particle, with electrostatic charges derived by the Discrete Surface Charge Optimization algorithm (DiSCO) (*Beard and Schlick, 2001b*); linker DNA, modeled with a combined worm-like chain and bead model (*Jian et al., 1997*); flexible histone tails, coarse-grained as five residues per bead (*Arya et al., 2006*); and linker histones (LHs) H1e and H1c, with coarse-grained beads for the globular domain (6 beads) and for the C-terminal domain (22 beads for H1e and 21 for H1c) (*Luque et al., 2014*; *Perišić et al., 2019*). Acetylated tails are modeled following our multiscale study on histone acetylation (*Collepardo-Guevara et al., 2015*). There, we showed that acetylated tails are more rigid and folded than WT tails, and that the chromatin unfolding upon acetylation occurs due to the impairment of internucleosome interactions caused by the folded and rigid tails. Thus, we use the configuration of folded tails and ensure rigidity by increasing the force constants in the energy terms by a factor of 100. $Mg^{2+}$ presence is modeled by a phenomenological approach (*Grigoryev et al., 2009*), in which the DNA persistence length is reduced from 50 nm to 30 nm based on experimental data (*Baumann et al., 1997*) and the electrostatic repulsion among linker DNAs is reduced by increasing the inverse Debye length in the DNA-DNA electrostatic term.

Coarse-grained elements have bonded interactions, which consist of stretching, bending, and twisting terms. Nonbonded interactions among coarse-grained elements are modeled with the Debye-Hückel approximation to treat the electrostatics and with Lennard-Jones potentials to treat excluded volume terms. For details on model parameters and energy terms, please see *Arya and Schlick, 2009*.

### Chromatin systems

To study quiescent and Log cell chromatin, we model a segment of 40 kb located between 130,000 bp and 170,000 bp of Chr1 (*Figure 2—figure supplement 2A, B*). This segment was selected based on Micro-C contact maps that show significant chromatin reorganization in this region, although the differences between Log and quiescence persist throughout the genome. Moreover, this region does not show a strong presence of condensin binding (*Swygert et al., 2019*). Nucleosome positions were obtained from MNase-seq data (*McKnight et al., 2015*) using the DANPOS algorithm (*Chen et al., 2013*). Nucleosomes called by DANPOS with summit values below 1% of the average summit value per condition were removed. Linker histone and histone acetylation positions were obtained from ChIP-seq data using using the 'callpeak' function of the MACS2 algorithm (*Zhang et al., 2008*). As a result, the Log chromatin fiber contains 222 nucleosomes, 61 nucleosomes acetylated, and an LH density of 0.05 LH/nucleosome. The quiescent chromatin fiber contains 228 nucleosomes, 3 nucleosomes acetylated, and an LH density of 0.29 LH/nucleosome (*Figure 2—figure supplement 2A, B*). The specific parameters used can be found in *Figure 2—figure supplement 2—source data 1*.

## Monte carlo sampling

Fibers representative of Log and quiescent cells are subject to Monte Carlo (MC) simulations starting from an ideal zigzag geometry as we have shown this configuration to be dominant under physiological salt conditions (**Grigoryev et al., 2009**). 50 independent trajectories are run for each system for at least 60 million MC steps. Each simulation is initiated from a different random seed number and a randomly chosen B twist value for the DNA of –12°, 0°, or +12° to mimic natural variations (**Drew and Travers, 1985**). To mimic physiological conditions, simulations are performed in the presence of 150 mM NaCl and 1mM Mg$^{2+}$, and a temperature of 293K.

Five types of tailored MC moves are implemented for the efficient global and local sampling of the fibers. A global pivot move chooses a random position along the fiber and then rotates the shorter section of the bisected chain around a randomly chosen axis running through that point. The resulting configuration is accepted or rejected based on the Metropolis criteria (**Metropolis and Ulam, 1949**). All DNA and LH beads are subject to translation and rotation moves also accepted or rejected based on the Metropolis criteria. A configurationally biased regrow routine is used to simulate the rapid movement of histone tails. A randomly chosen histone tail is regrown starting with the bead closest to the core; the new configuration is accepted or rejected based on the Rosenbluth criteria (**Rosenbluth and Rosenbluth, 1955**). Acetylated tails are sampled with a fold-swap move in which tails are randomly chosen and its fold state is swapped. Thus, if a chosen tail is currently folded (acetylated), its coordinates and equilibrium values are swapped with those of the unfolded version of that tail (WT). The new configuration is accepted or rejected based on the Metropolis criteria. Folded tails do not interact with any other chromatin element and are not subject to the regrow routine.

During the MC simulation, convergence of the systems is carefully checked by monitoring global and local properties (**Figure 2—figure supplement 3A, B**). The last 10 million MC steps of each independent trajectory, corresponding to a total of 5000 configurations, are used for analysis.

## Analysis

Raw data for each of the following analysis parameters are provided in **Figure 2—figure supplement 3—source data 1**. The sedimentation coefficient ($S_{w,20}$), in units of Svedbergs, is used to describe the compaction of the fiber. It is defined by the expression:

$$S_{w,20} = \left( (S_1 - S_0) * LH_{conc} + S_0 \right) * \left( 1 + \left( \frac{R_1}{N_C} \right) \sum_i \sum_j \frac{1}{R_{ij}} \right),$$

where $S_0$ is the sedimentation coefficient for a mononucleosome with LH bound ( 12S) (**Butler and Thomas, 1998**), $S_1$ is the sedimentation coefficient for a mononucleosome without LH ( 11.1S) (**Garcia-Ramirez et al., 1992**), $LH_{conc}$ is the concentration of LH in the fiber, $R_1$ is the spherical radius of a nucleosome (5.5 nm), $N_C$ is the number of nucleosomes in the fiber, and $R_{ij}$ is the distance between two nucleosomes i and j.

The radius of gyration, which describes the overall dimension of the polymer chain, is measured as the root mean squared distance of each nucleosome from the center of mass according to the relation:

$$\frac{1}{N_C} \sum_{j=1}^{N} (r_j - r_{mean})^2,$$

where $N_c$ is the number of nucleosomes, $r_j$ is the center position of the nucleosome core j, and $r_{mean}$ is the average of all core positions (**Perišić et al., 2010**).

Fiber volumes are calculated using the AlphaShape function of Matlab, which creates a nonconvex bounding volume that envelops the nucleosomes. Surfaces are visually inspected to ensure that they represent correctly the fiber morphology (**Figure 2—figure supplement 3C**). This is because noncylindrical-like shapes may not well be estimated. In that case, the AlphaShape object can be manipulated to tighten or loosen the fit around the points to create a nonconvex region.

Packing ratio is used to describe the compaction of the fiber and is measured as the number of nucleosomes contained in 11 nm of fiber. It is determined according to the relation:

$$Packing\,ratio = \frac{11 \cdot N_C}{fiber_{length}}$$

where $N_C$ is the number of nucleosomes and the fiber length is calculated using a cubic smoothing spline function native from Matlab.

## Nucleosome clustering interactions

We quantify the nucleosome clusters in each fiber by calculating the average number of nucleosomes per cluster, maximum number of nucleosomes per cluster, and average number of clusters. We use the density-based clustering algorithm DBSCAN (*Ester et al., 1996*), as implemented in MATLAB. DBSCAN partitions the n-by-n internucleosome distance matrix into clusters based on the neighborhood search radius (20 nm) and a minimum number of neighbors in the given neighborhood (three nucleosomes). The parameters are calculated for each of the 5000 configurations, and average and standard deviations are determined.

## Internucleosome interactions

Internucleosome contacts were calculated and reported every 10,000 steps during the simulation. A contact is defined if the tails or charge beads of nucleosome i are found to be within 2 nm of the tails or charge beads of nucleosome j. Contact maps for each trajectory are normalized by the maximal number of contacts seen throughout the trajectory, and the resulting normalized frequencies are summed together.

These internucleosome matrices are projected into normalized one-dimensional plots that depict the relative intensity of interactions between nucleosomes separated by k neighbors as follows:

$$I\left(k\right) = \frac{\sum_{i=1}^{N_C} I'\left(i, i\pm k\right)}{\sum_{j=1}^{N_C} I(j)}$$

where Nc is the number of nucleosomes and I' is the internucleosome interaction matrix.

## Tail interactions

For each of the 50 independent trajectories, containing about 600 chromatin configurations, we measure the fraction of configurations in which each tail $t$ ($t$=H2 A-N, H2A-C, H2B, H3, and H4) is 'in contact' with a chromatin element $e$ ($e$=separate core, separate DNA, or tails of a separate nucleosome). Namely, number of occurrences divided by the total sampled tail configurations. Note that, unlike the other analyses in the rest of this work, in this analysis and the analysis in the 'internucleosome interactions,' we include all 60 million MC steps, saving the configurations every 100,000 steps, resulting in 600 configurations.

Thus, we can construct a two-dimensional matrix, where each matrix element $T$ is defined as:

$$T'_{(t,e)} = mean\left[\frac{1}{N_C N_e} \sum_{i \in I_C} \sum_{j_e=1}^{N_e} \delta_{i,j_e}^{t,e}\left(M\right)\right]$$

with

$$\delta_{i,j_e}^{t,e}\left(M\right) = \begin{cases} 1 & if\ contact \\ 0 & otherwise \end{cases}$$

where $N_C$ is the total number of nucleosomes, $N_e$ is the number chromatin of elements, $I_C$ is a specific nucleosome along the chromatin fiber, and $M$ is a specific chromatin configuration. Thus, $\delta_{i,j_e}^{t,e}\left(M\right) = 1$ if $j_e$ is a e-type element 'in contact' with a $t$-type tail of nucleosome   at configuration $M$. We then define

$$T_{(t,e)} \frac{T'_{(t,e)}}{\sum_{e'=1}^{N_e} T'_{(t,e')}}$$

as the normalized tail interactions. The normalized tail interactions are reported in *Figure 2E* and *Figure 2—figure supplement 3F, G*.

For a given configuration (M), we consider a specific $t$-kind tail of nucleosome   to be either free or in contact with only one of the $N_e$ chromatin elements. Thus, the $t$-tail is in contact with an element $e$ if the shortest distance between its beads and the beads of $e$ is smaller than the shortest distance to

any other element *e* and also smaller than 2 nm. This distance criterion (2 nm) ensures that only tail beads that are strongly attracted to other chromatin elements are counted. The resulting interaction patterns provide insights into the frequency by which each *t*-kind tail mediates chromatin interactions.

The interaction frequencies are averaged over the 50 trajectories for each system, obtaining means and standard deviations. Raw data for tail interactions can be found in *Figure 2—source data 1*, *Figure 2—figure supplement 3—source data 2Figure 2—figure supplement 3—source data 3*.

## Micro-C XL data analysis

### Sequencing read processing

The two ends of paired-end Micro-C reads were mapped independently to the sacCer3 reference genome (release R64-2-1) using bowtie2 version 2.3.5.1 with the '--very-sensitive' parameter set (*Langmead and Salzberg, 2012*). All read pairs where either end received MAPQ score <6 were removed. All the remaining in-facing read pairs were removed. The resulting read pairs were processed into the multiple-bin size contact matrix in the Cooler format (https://github.com/mirnylab/cooler; *Abdennur and Mirny, 2020*). The bin sizes we used in the downstream analyses were 10 bp, 200 bp, and 5000 bp. Micro-C heatmaps were generated using Juicebox (*Durand et al., 2016*).

### Contact probability decay curve

Each diagonal of the 10 bp Cooler contact matrix contains the Micro-C interactions between genomic loci at the genomic distance of a multiple of 10 bp. The contact counts in each diagonal were summed and the sum was then divided by the number of elements in the diagonal. The result is then normalized so that the contact probability decay sums to one across all genomic distances. Analysis was completed separately for each orientation of ligation pairs ('in,' 'out,' and 'same,' which includes 'in-out' and 'out-in' pairs). 'Same' contacts are shown in all figures as these showed the most consistent nucleosome phasing across samples. The cumulative sum of the contact probability within a genomic distance range is reported as the cumulative contact probability. The sum of contact probability between 50 bp and 500 bp is called the short-range contact probability and that between 500 bp and 1000 bp is called the long-range contact probability. The ratio between the long-range and short-range contact probabilities is called the contact probability odd ratio.

### Micro-C contacts pileup analysis

We used the Micro-C contact matrix of 200 bp bin size to perform pileup analysis. Previous (*Swygert et al., 2019*) work defined the condensin ChIP peaks and L-CID boundaries (L-CID boundaries were previously called using the cworld-dekker package using matrix2insulation.pl with settings–is4800–nt0.4–ids3200–ss800–im mean; *Giorgetti et al., 2016*). A 20 kb window centering on each of the condensin ChIP peaks/L-CID boundaries is defined as a target region in this analysis. The set of Micro-C contacts between two target regions is defined as the submatrix of contacts between the two target regions. We call it an inter-peak submatrix if the two target regions correspond to two different condensin ChIP peaks/L-CID boundaries, or an intra-peak submatrix if the two target regions correspond to the same condensin ChIP peak/L-CID boundary. The element-wise median across a set of submatrices is defined as the pileup matrix. We call it an inter-peak pileup matrix if all the submatrices involved are inter-peak submatrices, or an intra-peak pileup matrix if all the submatrices are intra-peak submatrices. We also provide the element-wise mean across a set of submatrices in the figure supplements.

### HiCRep analysis

We implemented the HiCRep algorithm (*Yang et al., 2017*) in Python (*Lin et al., 2021*) to compute the SCC between two Micro-C contact matrices. The 5000 bp bin size contact matrices are used in this analysis. The input contact matrix is first normalized by dividing the contact counts by the sum of all contacts in the matrix. Then the chromosome-wise SCC scores between two normalized matrices are computed using the contacts up to 100 kb of genomic distance. The median of the chromosome SCC scores between the two matrices is reported.

## ChIP-seq data analysis

Reads were aligned to the sacCer3 reference genome (release R64-2-1) using bowtie2 in '--very-sensitive' mode (*Langmead and Salzberg, 2012*), then filtered and indexed using SAMtools (*Li et al., 2009*). Bam files were then RPKM normalized and converted to bigwig files using the 'bamCompare' command in deepTools3.0 (*Ramírez et al., 2016*), and IPs were normalized to inputs using 'bigwig-Compare.' Bigwigs were annotated using the list of Pol II transcript sites from *Pelechano et al., 2013*. Hierarchical clustering of Pol II ChIP-seq was completed using the 'hclust 2' command. Heatmaps, metaplots, and Pearson correlations were also generated using deepTools. Genome browser views were generated using the Integrated Genome Browser software (*Freese et al., 2016*). Peak calling was completed using 'callpeak' and 'bdgdiff' commands in MACS2 (*Zhang et al., 2008*). For final analysis, fastq files from two biological replicates were merged for each condition.

## Materials availability

All plasmids and yeast strains are available on request.

## Data and code availability

Genomics data are publicly available for download from NCBI GEO, accession number GSE167020. Genomics and mesoscale modeling analysis scripts are publicly available on GitHub at the following address: https://github.com/sswygert/Local-Chromatin-Fiber-Folding-Represses-Transcription-and-Loop-Extrusion-in-Quiescent-Cells (copy archived at swh:1:rev:c55fe78d7b99ce7ed-9d3396a619a4a841aecce0c, *Ledesma, 2021*).

## Acknowledgements

The authors would like to thank the past and current members of the Tsukiyama lab for their assistance and comments, especially Christine Cucinotta, Tianhong Fu, Laura Hsieh, and Jeffrey McKnight. The authors are grateful to our colleague Harmit Malik for his always insightful comments. The authors would also like to thank the Fred Hutch Genomics shared resource. Some images were created with BioRender.com.

## Additional information

### Funding

| Funder | Grant reference number | Author |
|---|---|---|
| National Cancer Institute | T32CA009657 | Sarah G Swygert |
| National Institute of General Medical Sciences | F32GM120962 | Sarah G Swygert |
| National Institute of General Medical Sciences | K99GM134150 | Sarah G Swygert |
| Academia Sinica | AS-CFII-108-119 | Po-Yen Lin |
| National Institute of General Medical Sciences | R01GM055264 | Tamar Schlick |
| National Institute of General Medical Sciences | R35GM122562 | Tamar Schlick |
| National Science Foundation | 2030277 | Tamar Schlick |
| National Institute of Diabetes and Digestive and Kidney Diseases | U54DK107979 | William S Noble |
| National Institute of General Medical Sciences | R01GM111428 | Toshio Tsukiyama |

| Funder | Grant reference number | Author |
| --- | --- | --- |
| National Institute of General Medical Sciences | R35GM139429 | Toshio Tsukiyama |
| Philip Morris International Inc. | | Tamar Schlick |
| Philip Morris USA Inc | | Tamar Schlick |

The funders had no role in study design, data collection and interpretation, or the decision to submit the work for publication.

## Author contributions

Sarah G Swygert, Conceptualization, Data curation, Formal analysis, Methodology, Methodology, Methodology, Project administration, Validation, Validation, Visualization, Writing – review and editing, Writing – review and editing; Dejun Lin, Data curation, Formal analysis, Methodology, Visualization, Writing – review and editing, Methodology, Software, Validation, ; Stephanie Portillo-Ledesma, Data curation, Formal analysis, Methodology, Methodology, Visualization, Writing – review and editing, Writing – review and editing, Software; Po-Yen Lin, Data curation, Formal analysis, Methodology, Visualization, Writing – review and editing, Funding acquisition, Methodology, Writing – review and editing; Dakota R Hunt, Data curation, Methodology, Formal analysis, Methodology, Writing – review and editing, Resources, Visualization; Cheng-Fu Kao, Supervision, Conceptualization, Investigation, Methodology, Writing – review and editing; Tamar Schlick, Methodology, Validation, Supervision, Conceptualization, Methodology, Software, Writing – review and editing; William S Noble, Methodology, Supervision, Conceptualization, Methodology, Writing – review and editing; Toshio Tsukiyama, Conceptualization, Methodology, Methodology, Project administration, Validation, Supervision, Writing – review and editing

## Author ORCIDs

Sarah G Swygert ⓘ http://orcid.org/0000-0002-0778-4624
Toshio Tsukiyama ⓘ http://orcid.org/0000-0001-6478-6207

## Decision letter and Author response

Decision letter https://doi.org/10.7554/eLife.72062.sa1
Author response https://doi.org/10.7554/eLife.72062.sa2

# Additional files

## Supplementary files

• Supplementary file 1. Statistical values for all chromatin volume measurements, including n, mean, and standard deviation for all replicates as well as p values of comparisons between samples.

• Transparent reporting form

## Data availability

All genomics data have been deposited to GEO and are available under accession code GSE167020. Genomics and mesoscale modeling analysis scripts are publicly available on GitHub at the following address: https://github.com/sswygert/Local-Chromatin-Fiber-Folding-Represses-Transcription-and-Loop-Extrusion-in-Quiescent-Cells (copy archived at https://archive.softwareheritage.org/swh:1:rev:c55fe78d7b99ce7ed9d3396a619a4a841aecce0c).

The following previously published datasets were used:

| Author(s) | Year | Dataset title | Dataset URL | Database and Identifier |
| --- | --- | --- | --- | --- |
| McKnight JN | 2015 | Rpd3 drives transcriptional quiescence | https://www.ncbi.nlm.nih.gov/geo/query/acc.cgi?acc=GSE67151 | NCBI Gene Expression Omnibus, GSE67151 |

*Continued*

| Author(s) | Year | Dataset title | Dataset URL | Database and Identifier |
|-----------|------|---------------|-------------|-------------------------|
| Swygert SG | 2018 | Condensin-dependent chromatin condensation represses transcription globally during quiescence | https://www.ncbi.nlm.nih.gov/geo/query/acc.cgi?acc=GSE120606 | NCBI Gene Expression Omnibus, GSE120606 |

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
