## [Editor Report]

The authors provide compelling evidence that the repression of gene expression during quiescence of the model eukaryote yeast is achieved by heterogenous clustering of local groups of nucleosomes.

---

## [Decision Letter]

**Decision letter after peer review:**

Thank you for submitting your article "Local Chromatin Fiber Folding Represses Transcription and Loop Extrusion in Quiescent Cells" for consideration by *eLife*. Your article has been reviewed by 2 peer reviewers, and the evaluation has been overseen by a Reviewing Editor and Kevin Struhl as the Senior Editor. The following individual involved in review of your submission has agreed to reveal their identity: Vijay Ramani (Reviewer #2).

As you will see from the reviews, the reviewers were generally enthusiastic about the work but also agreed that additional clarifications on data analysis and interpretation were required. The reviewers have discussed their reviews with one another, and the Reviewing Editor has drafted this list of essential revisions to help you prepare a revised submission.

Essential revisions:

1) The authors use the words long- range and neighboring-interactions without clearly defining them. It appears that the term "long-range" applies to interactions between nucleosomes greater than N+1, but for most readers, this term would mean interactions kilobases apart. So, we ask the authors to clearly define the term "long-range". Once defined authors should discuss the changes (or lack of) in the long-range (kilobases) interactions.

2) The authors suggest that Q cells have more long-range interactions than log cells. However, in figure 1 supplement 2 B and D, there appear more distal interactions in log for Q at 1 KB resolution.

3) The authors should provide decay curves so the reader can understand better the range of interactions that are enriched or depleted.

4) The authors shift from G1 to log phase between figures. The authors need to justify these switches.

5) The tomograph image looks compelling in figure 2A but quantitation in figure 2C seems to make the difference non-existent. It is hard to believe the differences between the orange and blue bars are statistically significant. Unless the authors provide more compelling evidence for the statistical differences between Q and log cells by this analysis, the tomograph figure should be removed.

6) The conclusions from modeling are only relevant if the results are robust to small changes in the parameters. If small changes in values of the authors' parameters cause the differences between log and Q to disappear or reverse, the authors would have to justify their values for their parameters. Indeed, the robustness of the modeling to support the previous conclusions is not evident in Figure 2 supplement 1 E. The error bars are huge. Unless the authors provide more compelling evidence for the statistical differences in the model's predictions, the modeling figure should be removed. If the authors can address this concern, then the modeling paragraph will be greatly improved by a brief description of the rationale and the results of the simulation for readers that are not familiar with these mathematical models.

7) The use of non-parental nucleosomes is confusing. Non-parental usually means new nucleosomes incorporated during S phase. Are the authors are referring to cis (within a nucleosome) vs trans (between nucleosomes) interactions? Please clarify.

8) Figure 3 C should show log-phase cells for comparison to allow the reader to assess how intermediate is the change in nucleosome interactions induced by TSA.

9) The authors should provide an explanation or comment why 20% of acetylation defective cells can enter Q. Unaddressed it would imply that a significant fraction of cells can enter quiescence despite being transcriptional active. Is this an artifact reflecting their limited ability to purify Q cells?

10) Figure 4F should also have log to provide a baseline for loss of compaction in mutants

11) Line 1687: Use of the term epistatic seems incorrectly applied. It is not clear how activation masks compaction state. Do the authors mean activation is uncorrelated with the level of compaction? Please clarify.

12) Why are the stripes present in figure 6A (in the mutant R17R19A), not present in the same mutants in the previous micro-C maps at the same resolution (4d and Figure 5 figure supplement 1 d?

13) Discussion should be shortened. The paragraph on cation role in condensation has seems to not be directly related with the results presented in the paper. The connection should be made clear or the paragraph should be deleted.

14) The authors consistently refer back to the notion of 10 nm and 30 nm fibers throughout the text (e.g. the paragraph beginning on p.69; line 1583), and the intent of this seems to be to liken the compacted states observed here to the "elusive" 30 nm fiber (e.g. by noting that H4K16ac disrupts the 30 nm fiber in vitro). This represents a fundamental weakness of the paper, because while the evidence provided certainly demonstrate that a novel compacted state exists in quiescent cells, the data are also fairly convincing that this is not a 30 nm fiber (as evidenced by STEM tomogram comparisons), especially given the bulk-averaged nature of the majority of the datasets presented here. Figure 1A illustrates precisely how the Hi-C data as presented can provide a false impression: contact curves demonstrating a typical contact probability decay are positioned under an individual fiber; these Micro-C data, however, are derived from many dinucleosome ligations averaged over many fibers from a large number of haploid cells, and thus cannot really be used to make statements about the structures of individual chromatin fibers. This issue does not dramatically alter the conclusions of the study, but does require a text revision of the 30 nm comparisons used throughout the paper (and, preferably, a sentence addressing the population averaged nature of Hi-C data).

15) Generally, the paper could be improved by more rigorous quantitative analysis of the Hi-C data, as most of the data is presented qualitatively in the current manuscript. Hi-CRep is useful for measuring reproducibility of replicates, but unfortunately remains the only significant quantitative comparison made across the many information-rich Micro-C XL datasets analyzed for this paper. This would be especially important for the contact decay curves presented – as of now it is very difficult for the reader to gauge how large a quantitative effect the various perturbations presented here have across their various samples. Moreover, it seems likely that the genome-wide averages presented here end up averaging over regions where there are larger or smaller reproducible changes in contact decay. We suggest that the authors formalize a way to quantify the relative change in contact probabilities (perhaps, through a log-odds ratio), but also consider performing these analyses across loci of interest. This extends to analyses presented in e.g. Figure 6 – can the authors use aggregate peak analysis of similar to provide some quantification of the heatmaps presented in Figure 6C?

16) The DSG results are somewhat odd and run counter to what one would expect from work by the Dekker, Rando, and Tjian-Darzacq labs in both yeast and mammalian cells. Is it possible that these findings underly a technical challenge in working with quiescent cells? One could imagine that e.g. differential accessibility to the DSG crosslinker could explain how the addition of DSG has minor effects on genome-wide contact probability estimates. Aside from testing MNase digestion efficiency by targeting ~95% mononucleosomes, the authors should mention if they considered any other ways to ensure that the contact probability decay curves being observed are not partially biased by differential crosslinking (or differential in situ ligation efficiency) in quiescent cells.

17) It would be great for the authors to make their analysis code / notebooks available via GitHub. This includes both the genomic data analysis code, the mesoscale modeling code, and any scripts used for analyzing the STEM tomograms.

---

## [Author Response]

Essential revisions:1) The authors use the words long- range and neighboring-interactions without clearly defining them. It appears that the term "long-range" applies to interactions between nucleosomes greater than N+1, but for most readers, this term would mean interactions kilobases apart. So, we ask the authors to clearly define the term "long-range". Once defined authors should discuss the changes (or lack of) in the long-range (kilobases) interactions.

We have updated the text to define “short-range” as less than 500 bp and “long-range” as between 500-1000 bp (for example, lines 149-150). We also provide odds ratios showing the ratio between contacts at these distances in Figure 1A, Figure 1**–**Supplement 3E, Figure 3C, Figure 4F, and Figure 7D. We also define longer than several kb contacts as “distal” and discuss differences in these contacts when appropriate throughout the text (for example, line 142). These additions greatly improve the text and figures and we are grateful for the excellent suggestion.

2) The authors suggest that Q cells have more long-range interactions than log cells. However, in figure 1 supplement 2 B and D, there appear more distal interactions in log for Q at 1 KB resolution.

The differences between distal contacts in log and Q are difficult to eyeball in the previous Figure 1 Supplement 1B and D (now Figure 1**–**Supplement 3B and D). Instead, we direct the reviewers to the metaplots in Figure 1D, in which log and Q data are shown up to 10 kb and a subtraction plot is provided showing there are more contacts in Q in this range. We have also added a cumulative contact probability plot in Figure 1**–**Supplement 1A showing contacts from 10 kb-1 mb. These show that contacts are higher in Q than log at the hundreds of kb to megabase range. Interestingly, there is a range, around 50-100 kb, at which log contacts are higher than Q. This is likely due to the formation of condensin-dependent loop domains that become insulated from each other due to condensin loop extrusion during Q. This difference between contacts in log and Q at the 1 kb range and above (largely mediated by condensin) is extensively covered in our previous paper, Swygert, *et al.*, *Molecular Cell*, 2019.

3) The authors should provide decay curves so the reader can understand better the range of interactions that are enriched or depleted.

In addition to the less than 1 kb contact decay curves in Figure 1A, Figure 1**–**Supplement 3E, Figure 3C, Figure 4F, and Figure 7D, we have added cumulative contact probability curves showing interactions between 10 kb to 1000 kb in Figure 1**–**Supplement 1A and Figure 4**–**Supplement 2A.

4) The authors shift from G1 to log phase between figures. The authors need to justify these switches.

We use log cells for Micro-C experiments to capture all major inter-nucleosomal contacts in actively dividing cells. Log is also the growth stage most commonly used in the field, simplifying comparisons between our quiescent cell results and existing data. For microscope work, we used G1 cells as a control so that we can compare the size of nuclei and chromatin fibers with the same DNA content as quiescent cells.

5) The tomograph image looks compelling in figure 2A but quantitation in figure 2C seems to make the difference non-existent. It is hard to believe the differences between the orange and blue bars are statistically significant. Unless the authors provide more compelling evidence for the statistical differences between Q and log cells by this analysis, the tomograph figure should be removed.

We have added mean values and 95% confidence intervals to the surface thickness quantifications in Figure 2A and added values of n to the Figure 2A legend. Although we agree that the difference between the mean fiber diameters in G1 and Q is relatively modest (~2 nm), the large number of diameters measured results in very high confidence in our calculated mean values.

We have also regrouped the fiber diameter distribution to 10 nm intervals, which is closer to the diameter of the nucleosome fiber, and removed the 0-5 nm group because it would be overestimated due to the distance between chromatin fibers in a crowded space. We also separated the histograms to make these differences more apparent by eye. The regrouped histograms show the major peak distribution of fibers in G1 cells is 5-10 nm and the population is also close to 50%, indicating that bare nucleosome fibers are dominant in chromatin organization. In contrast with G1 cells, the major peak distribution of quiescent cells is 10–20 nm, demonstrating that quiescent cell fibers appear more compact as compared to G1 cell fibers. As a positive control, the major two bin distributions are 10–20 nm and 20–30 nm in chicken erythrocyte nuclei.

To confirm this result, we have also estimated chromatin fiber diameters through an independent analysis of the STEM tomography data, added in Figure 2**–**Supplement 1. The new morphological erosion analysis shows that diameters in log are ~15% less than in Q, consistent with the surface thickness analysis in Figure 2.

The STEM tomography is an important complement to the Micro-C data, as it is able to determine the diameters of individual chromatin fibers, whereas Micro-C provides an average frequency of inter-nucleosomal interactions within the population of cells analyzed. Without these measurements, we would be unable to determine conclusively that quiescent cells do not contain 30 nm fibers.

6) The conclusions from modeling are only relevant if the results are robust to small changes in the parameters. If small changes in values of the authors' parameters cause the differences between log and Q to disappear or reverse, the authors would have to justify their values for their parameters. Indeed, the robustness of the modeling to support the previous conclusions is not evident in Figure 2 supplement 1 E. The error bars are huge. Unless the authors provide more compelling evidence for the statistical differences in the model's predictions, the modeling figure should be removed. If the authors can address this concern, then the modeling paragraph will be greatly improved by a brief description of the rationale and the results of the simulation for readers that are not familiar with these mathematical models.

We thank the reviewers for pointing out this issue. In rechecking the original data from a statistical ensemble of 30 trajectories, we discovered some data errors in transferring the files from the supercomputer to our local machines. When corrected, the data show very good trends. We now added another 20 trajectories to the ensemble (a total of 50 trajectories), and the results are shown in Figure 2E. As we see in the updated Figure 2E, the frequencies of interactions show a clear increase in Q chromatin versus Log chromatin, and the standard deviations are small. We have included the raw data for the tail interactions in Figure 2—source data 1, Figure 2—figure supplement 3—source data 2, and Figure 2—figure supplement 3—source data 3. Moreover, to prove that the two samples are statistically different, we performed an ANOVA test (*1*) and provided the p values for each pair of interactions. We also updated the methods section to include more detail about the tail interaction analysis. We also updated the compaction parameters presented in Figure 2-Supplement 3D using the larger ensemble constructed with the 50 independent trajectories.

We modified the paragraph in the main text based on these updated results, as well as to clarify that differences between the Log and Q chromatin fibers originate from differences in associated nucleosome positions, acetylation levels, and linker histone density. All these parameters were carefully taken from experimental data, such as MNase-seq and ChIP-seq. Thus, the differences observed in the compaction patterns and tail interactions arise from the combination of the parameters used to represent the two different fibers. As we have previously shown, fiber compaction and overall shape is affected by LH density (*2*–*4*), nucleosome positions (*2*, *5*, *6*), and histone tail acetylation (*2*, *7*–*9*), and the folding characteristics of genes depend on these parameters (*10*, *11*).

Moreover, we provided more details on the specific parameters used to construct both models, such as which nucleosomes have linker histone bound, which nucleosomes are acetylated, and the list of linker DNAs between each nucleosome. This information can be found in Figure 2—figure supplement 2—source data 1. We also include an image in Figure 2-Supplement 2B of the MNase-seq data and ChIP-seq data used to determine the parameters of each fiber (shown in Figure 2-Supplement 2A); it is evident that these data differ between Log and Q chromatin. Finally, we provide the raw data used to calculate the average and standard deviation for each parameter in Figure 2—figure supplement 3—source data 1.

1. J. Kaufmann, A. G. Schering, in *Wiley StatsRef: Statistics Reference Online* (American Cancer Society, 2014).

2. S. Portillo-Ledesma *et al.*, Nucleosome Clutches are Regulated by Chromatin Internal Parameters. *J. Mol. Biol.* 433, 166701 (2021).

3. N. Yusufova *et al.*, Histone H1 loss drives lymphoma by disrupting 3D chromatin architecture. *Nature* (2020), doi:10.1038/s41586-020-3017-y.

4. O. Perišić, S. Portillo-Ledesma, T. Schlick, Sensitive effect of linker histone binding mode and subtype on chromatin condensation. *Nucleic Acids Res.* 47, 4948–4957 (2019).

5. O. Perišić, R. Collepardo-Guevara, T. Schlick, Modeling Studies of Chromatin Fiber Structure as a Function of DNA Linker Length. *J. Mol. Biol.* 403, 777–802 (2010).

6. G. Bascom, T. Kim, T. Schlick, Kilobase Pair Chromatin Fiber Contacts Promoted by Living-System-Like DNA Linker Length Distributions and Nucleosome Depletion. *J. Phys. Chem. B*. 121, 3882–3894 (2017).

7. S. S. P. Rao *et al.*, Cohesin Loss Eliminates All Loop Domains. *Cell*. 171, 305–320 (2017).

8. R. Collepardo-Guevara *et al.*, Chromatin Unfolding by Epigenetic Modifications Explained by Dramatic Impairment of Internucleosome Interactions: A Multiscale Computational Study. *J. Am. Chem. Soc.* 137, 10205–10215 (2015).

9. G. D. Bascom, T. Schlick, Chromatin Fiber Folding Directed by Cooperative Histone Tail Acetylation and Linker Histone Binding. *Biophys. J.* 114, 2376–2385 (2018).

10. G. Bascom, C. Myers, T. Schlick, Mesoscale modeling reveals formation of an epigenetically driven hoxc gene hubs. *Proc. Natl. Acad. Sci. USA*. 116, 4955–4962 (2018).

11. P. A. Gómez-García *et al.*, Mesoscale Modeling and Single-Nucleosome Tracking Reveal Remodeling of Clutch Folding and Dynamics in Stem Cell Differentiation. *Cell Rep.* 34 (2021), doi:10.1016/j.celrep.2020.108614.

7) The use of non-parental nucleosomes is confusing. Non-parental usually means new nucleosomes incorporated during S phase. Are the authors are referring to cis (within a nucleosome) vs trans (between nucleosomes) interactions? Please clarify.

We agree with the reviewers’ comments and changed the text as suggested.

8) Figure 3 C should show log-phase cells for comparison to allow the reader to assess how intermediate is the change in nucleosome interactions induced by TSA.

This has been added as suggested.

9) The authors should provide an explanation or comment why 20% of acetylation defective cells can enter Q. Unaddressed it would imply that a significant fraction of cells can enter quiescence despite being transcriptional active. Is this an artifact reflecting their limited ability to purify Q cells?

We believe the reviewers are referring to quiescence entry of H4 basic patch mutants, which is up to ~80% less efficient than WT (Figure 4**–**Supplement 1B). We use Percoll density gradients to separate quiescent cells from non-quiescent cells in 7-day stationary culture, which is the most established method to purify quiescent cells (Allen et al., 2006). This method relies on the fact that quiescent cells become more dense during quiescence entry. Therefore, our results show that up to about 1/5th of H4 basic patch mutants, which are transcriptionally more active than WT cells, can still undergo the physiological changes that allow them be separated from non-quiescent cells on density gradients. Why not all mutant cells behave in the same way (e.g. zero % quiescence entry) is an interesting question that we would like to follow up on in a future work, but it is similar to the use of cells with growth defects in log conditions, as is commonly done in the field. In addition to the increase in density, purified mutant quiescent cells still exhibit longevity that is comparable to WT quiescent cells (Figure 4—figure supplement 1C), and although transcription is generally de-repressed, the mutant cells express quiescence-specific genes that are not expressed in log. Therefore, we believe it is adequate to call them mutant quiescent cells.

10) Figure 4F should also have log to provide a baseline for loss of compaction in mutants

This has been added as suggested.

11) Line 1687: Use of the term epistatic seems incorrectly applied. It is not clear how activation masks compaction state. Do the authors mean activation is uncorrelated with the level of compaction? Please clarify.

We agree and have changed the word from “epistatic” to “downstream”. We have shown that H4 mutation leads to less compaction even in genes that do not become activated, suggesting that loss of compaction is necessary but not sufficient for activation.

12) Why are the stripes present in figure 6A (in the mutant R17R19A), not present in the same mutants in the previous micro-C maps at the same resolution (4d and Figure 5 figure supplement 1 d?

Condensin/L-CID boundaries are not evenly distributed throughout the genome, and the region shown in Figure 6A does not have significant condensin binding. However, Figure 5**–**Supplement 1 shows stripes at ~72 and ~93 kb. Stripes are also clearly visible in the metaplots in Figure 4G, as these are centered on condensin-bound L-CID boundaries.

13) Discussion should be shortened. The paragraph on cation role in condensation has seems to not be directly related with the results presented in the paper. The connection should be made clear or the paragraph should be deleted.

We agree the Discussion was too long and have substantially shortened it.

14) The authors consistently refer back to the notion of 10 nm and 30 nm fibers throughout the text (e.g. the paragraph beginning on p.69; line 1583), and the intent of this seems to be to liken the compacted states observed here to the "elusive" 30 nm fiber (e.g. by noting that H4K16ac disrupts the 30 nm fiber in vitro). This represents a fundamental weakness of the paper, because while the evidence provided certainly demonstrate that a novel compacted state exists in quiescent cells, the data are also fairly convincing that this is not a 30 nm fiber (as evidenced by STEM tomogram comparisons), especially given the bulk-averaged nature of the majority of the datasets presented here. Figure 1A illustrates precisely how the Hi-C data as presented can provide a false impression: contact curves demonstrating a typical contact probability decay are positioned under an individual fiber; these Micro-C data, however, are derived from many dinucleosome ligations averaged over many fibers from a large number of haploid cells, and thus cannot really be used to make statements about the structures of individual chromatin fibers. This issue does not dramatically alter the conclusions of the study, but does require a text revision of the 30 nm comparisons used throughout the paper (and, preferably, a sentence addressing the population averaged nature of Hi-C data).

We did not intend to imply that the 30 nm fiber exists in quiescent cells. Indeed, we have repeatedly mentioned that the quiescence-specific compact, disordered local chromatin fiber folding is distinct from that of 30 nm fibers (for example, lines 202-205, 227-229). We agree with the reviewers that Micro-C provides average contact probability within the population. In contrast, our STEM tomography measures individual chromatin fibers (please see our response to comment 5 above). The Micro-C results clearly show that the pattern of contacts in quiescent cells is inconsistent with 30 nm folding, and the STEM results show that quiescent cell chromatin forms fibers much smaller in diameter than reconstituted 30 nm fibers or chicken erythrocyte fibers. To make this point more clear and avoid misunderstanding by readers, we have clarified this further in the text (for example, lines 281-283, 497-501).

15) Generally, the paper could be improved by more rigorous quantitative analysis of the Hi-C data, as most of the data is presented qualitatively in the current manuscript. Hi-CRep is useful for measuring reproducibility of replicates, but unfortunately remains the only significant quantitative comparison made across the many information-rich Micro-C XL datasets analyzed for this paper. This would be especially important for the contact decay curves presented – as of now it is very difficult for the reader to gauge how large a quantitative effect the various perturbations presented here have across their various samples. Moreover, it seems likely that the genome-wide averages presented here end up averaging over regions where there are larger or smaller reproducible changes in contact decay. We suggest that the authors formalize a way to quantify the relative change in contact probabilities (perhaps, through a log-odds ratio), but also consider performing these analyses across loci of interest. This extends to analyses presented in e.g. Figure 6 – can the authors use aggregate peak analysis of similar to provide some quantification of the heatmaps presented in Figure 6C?

We thank the reviewer for this constructive suggestion of quantifying the differences in different Micro-C XL data sets. Most of our current Micro-C analysis focuses on genomic distances corresponding to a few nucleosomes. The challenge of comparing the entire Micro-C XL map across data sets is the dominance of Rabl configuration contacts in the data, which makes it difficult to visualize the differences on the level of nucleosomal arrays. As the reviewer suggests here, we have decided to use the odds ratio between long (500 bp to 1000 bp) and short (50 bp to 500 bp) genomic distance contacts to highlight the differences across data sets. We have added these odds ratios as suggested to all contact decay curves (Figure 1A, Figure 1**–**Supplement 3E, Figure 3C, Figure 4F, and Figure 7D). We have also examined contact probabilities under 1 kb in differentially expressed gene clusters (added in Figure 1**–**Supplement 1B,C). Surprisingly, we see little difference in the contact patterns between expressed and unexpressed genes. In fact, we have been unable to find locations in which contacts seem to differ much from each other within a certain condition. This analysis and all heatmaps (for example, Figure 5**–**Supplement 1D-F) show that changes in contacts occur fairly uniformly across the genome in the conditions examined.

Figure 6C of our original submission was an aggregate analysis of the contacts between Brn1 ChIP-seq peaks. We presented the median of the contacts rather than the mean, which is typically used in aggregate analysis. The reason for using the median is to highlight the stripe-like interaction pattern of enriched interaction around the genomic distance of zero in Figure 6C. We also believe the median is a more accurate representation of the data than the mean, which can be thrown off by contact differences at a small number of loci. However, we understand that it’s important to provide the mean of the contacts and we have included these plots in Figure 6**–**Supplement 1D.

16) The DSG results are somewhat odd and run counter to what one would expect from work by the Dekker, Rando, and Tjian-Darzacq labs in both yeast and mammalian cells. Is it possible that these findings underly a technical challenge in working with quiescent cells? One could imagine that e.g. differential accessibility to the DSG crosslinker could explain how the addition of DSG has minor effects on genome-wide contact probability estimates. Aside from testing MNase digestion efficiency by targeting ~95% mononucleosomes, the authors should mention if they considered any other ways to ensure that the contact probability decay curves being observed are not partially biased by differential crosslinking (or differential in situ ligation efficiency) in quiescent cells.

Micro-C XL has not previously been performed in quiescent cells, and we believe the minor differences between DSG treated and untreated cells may reflect a decrease in chromatin dynamics or some other trait that is specific to quiescence (and possibly quiescent yeast). Although we believe it is unlikely, we cannot entirely rule out the possibility that this difference is that DSG works less efficiently in quiescent cells. However, if this is true, we are underestimating the increase in nucleosome contacts in quiescence. In this case, it would be most appropriate to compare the Log sample without DSG to either Q condition, which does not affect our interpretation of the results in any way.

17) It would be great for the authors to make their analysis code / notebooks available via GitHub. This includes both the genomic data analysis code, the mesoscale modeling code, and any scripts used for analyzing the STEM tomograms.

We have created a GitHub repository and added scripts for genomics analysis and mesoscale modeling as suggested, and have added the GitHub link to the manuscript. No scripts were used for the STEM analysis.